# Efficient and selective capture of thorium ions by a covalent organic framework

Xiaojuan Liu[1], Feng Gao[1], Tiantian Jin[1], Ke Ma[1], Haijiang Shi[1], Ming Wang[2], Yanan Gao[2], Wenjuan Xue[3], Jing Zhao [1] ✉, Songtao Xiao [1] ✉, Yinggen Ouyang [1] ✉ & Guoan Ye [1] ✉

The selective separation of thorium from rare earth elements and uranium is a critical part of the development and application of thorium nuclear energy in the future. To better understand the role of different N sites on the selective capture of Th(IV), we design an ionic COF named Py-TFImI-25 COF and its deionization analog named Py-TFIm-25 COF, both of which exhibit record-high separation factors ranging from $10^2$ to $10^5$. Py-TFIm-25 COF exhibits a significantly higher Th(IV) uptake capacity and adsorption rate than Py-TFImI-25 COF, which also outperforms the majority of previously reported adsorbents. The selective capture of Py-TFImI-25 COF and Py-TFIm-25 COF on thorium is via Th-N coordination interaction. The prioritization of Th(IV) binding at different N sites and the mechanism of selective coordination are then investigated. This work provides an in-depth insight into the relationship between structure and performance, which can provide positive feedback on the design of novel adsorbents for this field.

Due to the restriction of carbon emission, the proportion of fossil energy in the future energy structure will be gradually reduced. The most desired new energy source is nuclear energy since it offers a controlled reaction, a consistent energy output, and a high energy repayment ratio, and also is unrestricted by geography and the natural environment[1,2]. Thorium, in addition to uranium, is an important nuclear energy resource and a strategic reserve food for the nuclear industry's development[3–7]. Thorium ores are typically associated with rare earth and uranium elements[8–10]. Furthermore, a significant amount of thorium remains in waste tailings generated by the development of rare earth and uranium mines. As a result, the selective separation of thorium from rare-earth elements and uranium is a crucial part of the future development and application of thorium nuclear energy. However, the separation of thorium from rare-earth elements and uranium is quite challenging, because of their similar chemical properties[11,12].

Adsorption[13–15] is considered an effective method while compared to traditional methods such as solvent extraction[16,17] because of its simple operation, minimal secondary waste, and selectivity for target nuclides. The development of diverse adsorbent materials, such as clay minerals[18,19], zeolites[20–22], polymers[23–25], and metal-organic frameworks[26–31], for the capture of actinides, has thus been the focus of the research. The majority of these materials, however, also suffer from low iso-mass adsorption rates, high skeletal density, and poor selectivity. Consequently, developing novel materials with both superior adsorption selectivity and substantial entrapment capacity is an essential research area.

Covalent organic frameworks (COFs) with adjustable architectures and high chemical and thermal stability provide a unique and efficient platform for the separation of radionuclides with low concentrations[32–35]. In general, the capture of radionuclides by COF materials is based on the functional monomers in COFs to be distributed in specific spatial locations and to interact with radionuclides in a shape-selective[36–38] or complexation manner[14,39–45]. The complexation manner relies primarily on electron-rich atoms (e.g., nitrogen[39,40], oxygen[41], phosphorus[43], sulfur) to form charge-transfer complexes with electron-deficient actinide ions. However, oxygen sites usually have a high affinity for both lanthanide and actinide ions but

[1]Department of Radiochemistry, China Institute of Atomic Energy, 102413 Beijing, China. [2]Key Laboratory of Ministry of Education for Advanced Materials in Tropical Island Resources, Hainan University, 570228 Haikou, China. [3]State Key Laboratory of Separation Membranes and Membrane Processes, Tiangong University, 300387 Tianjin, China. ✉e-mail: Jing_zhao@tom.com; Songtao_Xiao@tom.com; ouyang_yinggen@163.com; guoan_ye@tom.com

poor selectivity. Materials containing sulfur or phosphorus sites may produce hazardous compounds to the environment during the post-treatment[46]. Based on the above considerations, we hypothesize that adsorbents containing N sites may exhibit promising affinity and selectivity for thorium ions.

To the best of our knowledge, there are only two cases that evaluated the capture of thorium by porous COFs. One is $[NH_4]^+[COF\text{-}SO_3^-]$ developed by Xiong et al.[45], the other is COF-DL229[47] reported by our group. $[NH_4]^+[COF\text{-}SO_3^-]$ containing sulfur sites exhibits the Th(IV) capture capacity of 395 mg g$^{-1}$, with separation factors demonstrated as $SF_{Th/U} = 6.9$, $SF_{Th/Eu} = 11.2$, and $SF_{Th/Ce} = 13.5$. COF-DL229 containing N sites shows a strong affinity towards Th(IV) with the saturated capacity of 500 mg g$^{-1}$, but little affinity with other cations. Therefore, to design and choose appropriate construction units, the nitrogen-containing elements seem to be more desirable. However, the effect of different N sites in COF materials on adsorption still needs to be explored. The removal performance of Th(IV) and competing ions are noticeably different, but the mechanism remains unclear. The electron-scale behavior and adsorption mechanism of Th(IV) and competing ions on COF materials need to be investigated in-depth.

In this work, we designed and synthesized an ionic COF named Py-TFImI-25 COF and its deionization analog named Py-TFIm-25 COF, to investigate the influence of N sites on the adsorption capacity and selectivity of Th(IV). The framework of Py-TFImI-25 COF was built through imine linkages generated by three monomers, 4,4′,4″,4‴-(Pyrene-1,3,6,8-tetrayl) tetraaniline (PyTTA), 2′3′5′6′-tetrafluoro-[1,1′:4′,1″-terphenyl]−4,4′-dicarbaldehyde (TFTDA), and 5,6-Bis(4-formylbenzyl)−1,3-dimethyl-benzimidazolinium iodide (BFIIm). Py-TFIm-25 COF is the deionized analog of the structure of Py-TFImI-25 COF, constructed from PyTTA, TFTDA, and 5,6-Bis(4-formylbenzyl)−1-methyl-1H-benzimidazole (BFIm) (see Fig. 1). The introduction of TFTDA is to enhance the molecular interlayer interaction between the fluorinated build block and non-fluorinated build block[48], which can improve the crystallinity and chemical stability of the COFs. The two COFs exhibited similar structure characteristics, being different only in N sites, allowing to investigate the role of N sites in the Th(IV) capture.

The capture properties of the two COFs for Th(IV) were assessed by investigating the optimum pH, the adsorption kinetics, and the selectivity for a direct benchmark with the adsorbents reported in the literature. Py-TFIm-25 COF exhibited an obviously higher Th(IV) uptake capacity and selectivity than Py-TFImI-25 COF, indicating that the type of N sites in the adsorbent plays a vital role in Th(IV) adsorption. The electronic localization function (ELF) contour maps combined with calculated adsorption potential energies revealed that Th(IV) binding affinity at different N sites follows the order $N_{Im} > N_{-C=N-} > N_{Im\text{-}CH_3}$. The multi-component adsorption performance revealed that both of the two COFs show a high affinity for Th(IV), but hardly adsorb other cations, suggesting good selectivity towards Th(IV). Furthermore, the mechanism of selective adsorption of Th(IV) was thoroughly explored through the density of states (DOS) and charge density difference.

## Results

### Characterization

Powder X-ray diffraction (PXRD) results revealed that both Py-TFImI-25 COF and Py-TFIm-25 COF were highly crystalline, exhibiting identifiable peaks at 2.64°, 3.78°, 8.06°, 10.78°, and 23.24° (Fig. 2a, c). To validate that the synthesized COFs had the desired structures, we implemented structural modeling based on PXRD. The results indicated that for both materials, the experimental data and the simulated data based on the eclipsed AA stacking model were in good agreement (Fig. 2b, d), and the observed diffraction peaks could be characterized as (110), (220), (330), (440), and (001) reflections, respectively. The Pawley-refined results produced the space group of P1 with a unit cell of $a = 43.30$ Å, $b = 7.94$ Å, $c = 50.22$ Å, $\alpha = 87.41°$, $\beta = 90.59°$, and

$\gamma = 81.35°$ ($R_P = 2.46\%$, $R_{WP} = 3.19\%$) for Py-TFImI-25 COF (Supplementary Table 1), and the space group of P1 with a unit cell of $a = 44.98$ Å, $b = 7.93$ Å, $c = 47.37$ Å, $\alpha = 88.52°$, $\beta = 90.90°$, and $\gamma = 85.17°$ ($R_P = 1.78\%$, $R_{WP} = 2.31\%$) for Py-TFIm-25 COF (Supplementary Table 2), confirming the correctness of peak assignments as evidenced by negligible deviation from the observed PXRD patterns. The interlayer distances of Py-TFImI-25 COF and Py-TFIm-25 COF were 4.5 Å and 4.1 Å, respectively. The cationic benzimidazolium linkers of Py-TFImI-25 COF implanted an ionic interface on the pore walls, preventing π−π stacking due to charge repulsion and resulting in the larger interlayer distance[49,50]. The Brunauer-Emmett-Teller (BET) surface areas of Py-TFImI-25 COF and Py-TFIm-25 COF were derived from the N$_2$ sorption isotherms collected at 77 K (Fig. 2e) and were 1324.05 and 1430.15 m$^2$ g$^{-1}$, respectively. The pore-size distributions of Py-TFImI-25 COF and Py-TFIm-25 COF were centered at 2.52 nm and 2.31 nm, respectively. The pore parameters suggested that both of these two COFs were predominantly mesoporous (Supplementary Table 3). In conclusion, the pore sizes of Py-TFIm-25 COF prepared by deionization modification and Py-TFImI-25 COF were nearly identical with no significant difference, indicating that the anion group has no impact on the size of the channel. As illustrated in Fig. 2b, d, it was due to the unique distribution of the iodide ions, which are tightly attached to the pore walls. Therefore, the deionization analog did not significantly change the pore volume.

The completion of the Schiff base reactions between PyTTA, TFTDA, and BFIIm/BFIm was demonstrated by the weakness of the aldehyde group band at 1698 cm$^{-1}$ and the synchronous appearance of the characteristic -C=N- stretching bands at 1624 cm$^{-1}$ in the Fourier transform infrared (FT-IR) spectra (Fig. 2f). Both COFs are layered in lamellar form, forming a film-like microcrystalline structure that was clearly visible under an electron microscope with an imaging resolution of 10 μm (Supplementary Fig. 14). Due to the fact that COFs are not perfect crystals by nature, the lamellar structure is not regular in all regions, and some of them are disordered, so the microstructures contain flawed parts. The thermal stability of both materials was tested under a nitrogen atmosphere (Supplementary Fig. 15). Both COFs were able to maintain more than 90% of their weight when the temperature was increased to 500 °C, and both materials retained more than 68% of their weight when the temperature raised to 1000 °C, indicating that the materials have outstanding thermal stability. The two materials exhibited good stability after soaking in water and maintained the integrity of the skeleton and the crystal structure (Supplementary Figs. 16 and 17).

### Capture of tetravalent thorium ions

To trap trace concentrations of thorium ions from rare-earth raffinate, the concentrations of thorium ions chosen for all the experiments were set as close as possible to that in the actual process[51]. The Th(IV) capture performance of Py-TFImI-25 COF and Py-TFIm-25 COF with different pH values was illustrated in Fig. 3a. The entrapment capacities of both materials were comparatively low between pH 1.0 to 2.0, and steadily rose between pH 3.0 to 5.0. The protonation of the N sites is the reason for the decreased adsorption capacity at low pH, and the pronated nitrogen atoms in functional groups are not capable to form complexes since there are no lone pair electrons available. As the pH value decreases, the majority of the adsorbents depending on coordination interaction would lose their adsorption capacity[52–54]. So, pH 4.0 was determined to be the optimal pH for subsequent adsorption experiments of Th(IV). The adsorption capacity of Py-TFIm-25 COF was higher than that of Py-TFImI-25 COF, indicating that Py-TFIm-25 COF had different capture sites and enhanced binding ability with thorium ions. The results showed that although Py-TFImI-25 COF and Py-TFIm-25 COF have analogous structures, the new N sites on the benzimidazole linkers of Py-TFIm-25 COF could be responsible for the improved sorption performance.

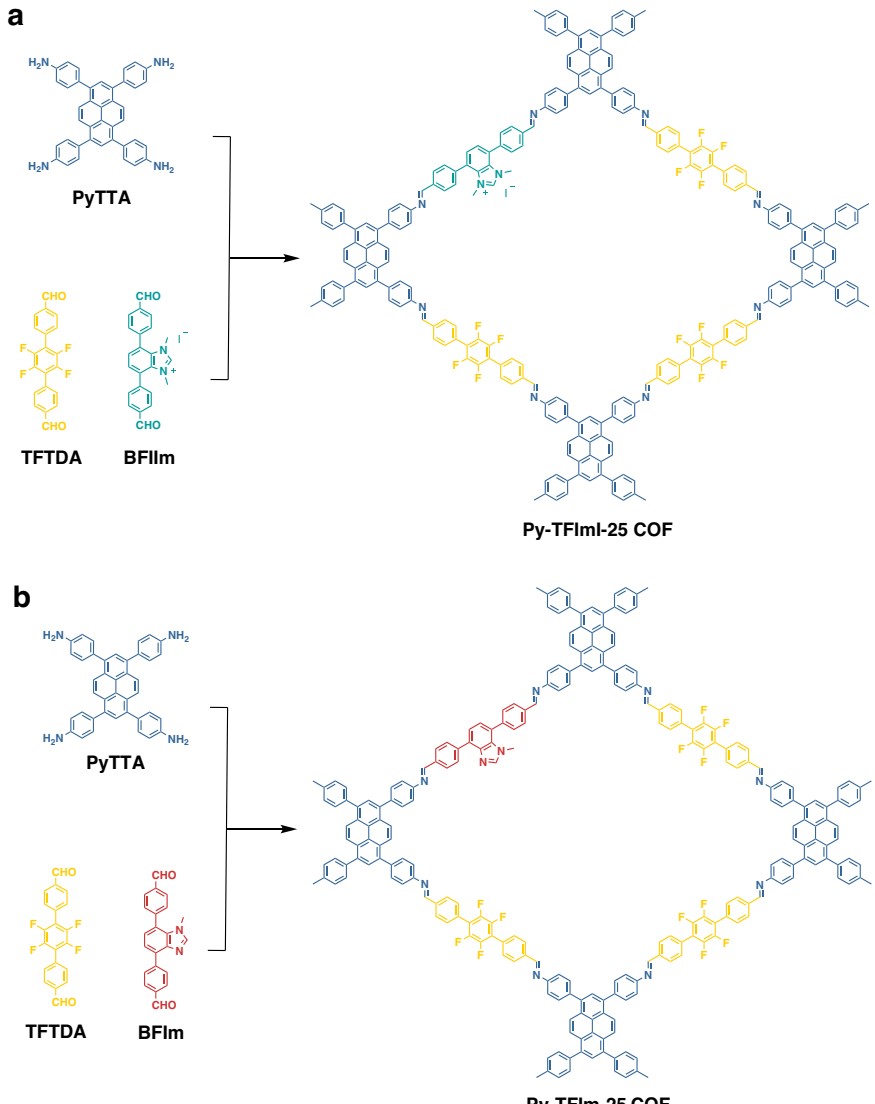

**Fig. 1 | Schematic of synthetic Py-TFImI-25 COF and Py-TFIm-25 COF. a** Synthesis of Py-TFImI-25 COF. **b** Synthesis of Py-TFImI-25 COF.

To investigate the capture efficiency of both COFs on Th(IV), the adsorption kinetics of Py-TFImI-25 COF and Py-TFIm-25 COF were performed. As shown in Fig. 3b, the adsorption rates of Th(IV) on the surfaces of both COFs were very rapid, with Py-TFImI-25 COF reaching equilibrium within 4 min (rate constant $k_2 = 4.52 \times 10^{-2}$ g mg$^{-1}$ min$^{-1}$, Supplementary Fig. 18), and with Py-TFIm-25 COF attaining equilibrium within 1 min. During the manual sampling approach, the rising phase of the Py-TFIm-25 COF adsorption capacity curve was not detected. Thus, developing a kinetic model for Py-TFIm-25 COF was not rigorous. However, the results confirmed that Py-TFIm-25 COF had higher adsorption capacity (Supplementary Fig. 19) and faster adsorption rate than Py-TFImI-25 COF, as well as higher adsorption efficiency than other reported adsorbent materials (Supplementary Figs. 20 and 21). The remarkable adsorption rates were attributed to the mesoporous structures that provide transparent one-dimensional channels for rapid access of thorium ions to the adsorption sites, as well as the high affinity of the adsorption sites for thorium ions.

The selectivity of Py-TFImI-25 COF and Py-TFIm-25 COF for Th(IV) in the presence of competing ions including U(VI), La(III), Pr(III), Nd(III), Sm(III), Eu(III), Gd(III), Sr(II), and Cs(I) was investigated as shown in Fig. 3c, d. As we expected, both materials demonstrated impressive selectivity towards Th(IV). Py-TFImI-25 COF exhibited a

distribution coefficient of $5.54 \times 10^3$ mL g$^{-1}$ for Th(IV), a distribution coefficient ranging from 0.41 to 37.70 mL g$^{-1}$ for lanthanide ions, 37.04 mL g$^{-1}$ for U(VI), 0.64 mL g$^{-1}$ for Sr(II), and 0.33 mL g$^{-1}$ for Cs(I). Meanwhile, Py-TFIm-25 COF exhibited a distribution coefficient of $1.21 \times 10^4$ mL g$^{-1}$ for Th(IV), a distribution coefficient ranging from 0.28 to 204.35 mL g$^{-1}$ for lanthanide ions, 30.80 mL g$^{-1}$ for U(VI), 3.18 mL g$^{-1}$ for Sr(II), and 0.33 mL g$^{-1}$ for Cs(I). The separation factors $SF_{Th(IV)/M}$ of the two materials for Th(IV) compared to other competing ions were all in the range of $10^2$ to $10^5$, which overwhelmed the majority of previously reported adsorbents[45,54,55]. These findings indicated that the affinity of Py-TFImI-25 COF and Py-TFIm-25 COF towards Th(IV) was considerably higher than that towards other competing cations. The selectivity of Py-TFIm-25 COF for Th(IV) was superior to Py-TFImI-25 COF due to its higher Th(IV) adsorption capacity.

## Discussion

Various characterization approaches were adopted to investigate the interaction mechanism of Py-TFImI-25 COF and Py-TFIm-25 COF towards Th(IV). The obtained PXRD patterns of Py-TFImI-25 COF and Py-TFIm-25 COF after Th(IV) adsorption exhibited consistent diffraction peaks with the pristine ones (Supplementary Fig. 22), indicating that the adsorption of Th(IV) would not disrupt the crystalline

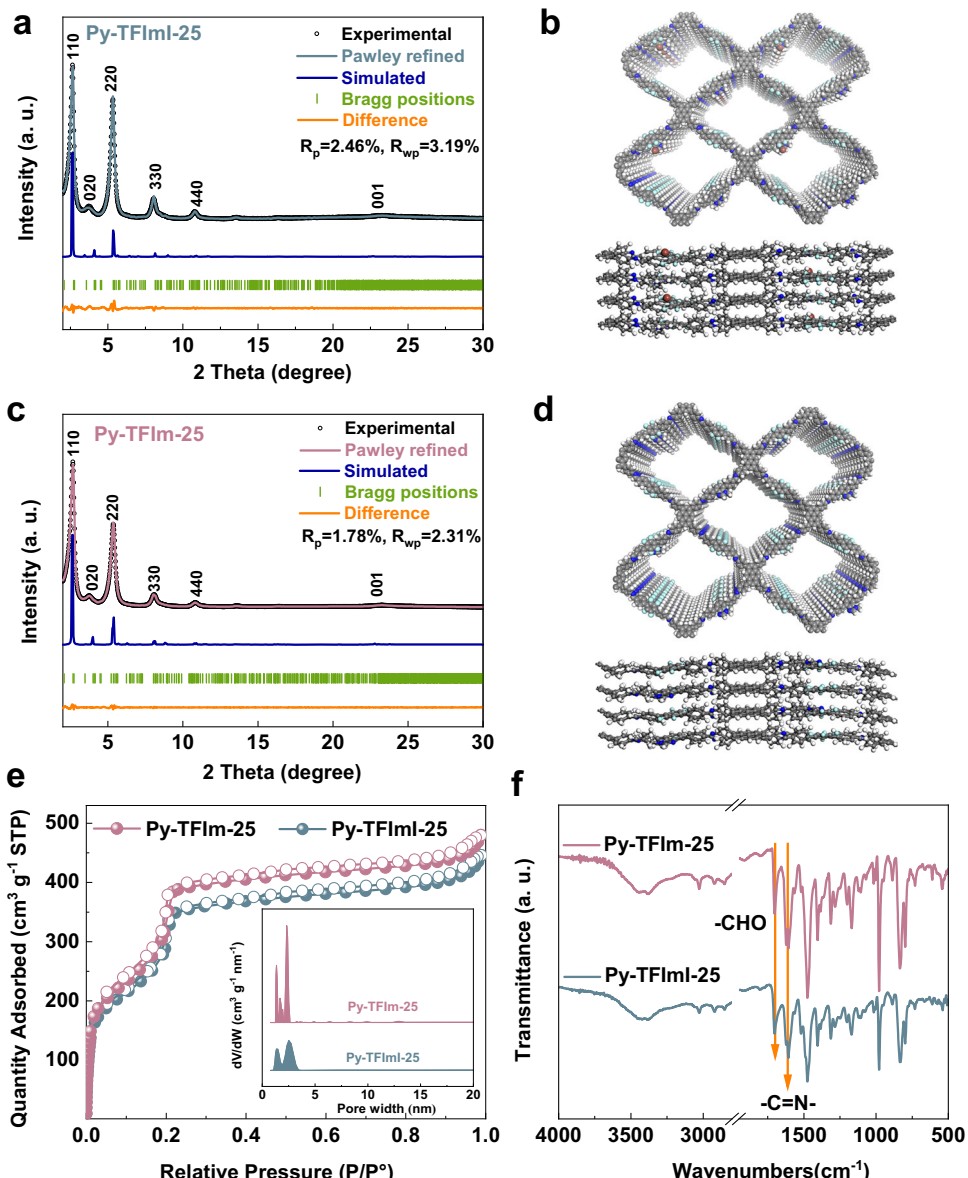

**Fig. 2 | Structural characterization of Py-TFImI-25 COF and Py-TFIm-25 COF.**
**a**, **c** PXRD profiles of **a** Py-TFImI-25 COF and **c** Py-TFIm-25 COF. **b**, **d** Structure models of **b** Py-TFImI-25 COF and **d** Py-TFIm-25 COF viewed along the c-axis (upper) and a-axis (lower) (gray, C; blue, N; light green, F; light gray, H; red, I). **e** N₂ sorption isotherms. The inset shows the pore-size distributions calculated from the non-local density theory. **f** FT-IR spectra.

structures. As shown in Fig. 4a, the characteristic peak of Th-N stretching vibration occurred in the FT-IR spectra of both materials after the adsorption of Th(IV) at 1384 cm⁻¹. The X-ray photoelectron spectroscopy (XPS) spectra in Fig. 4b showed the appearance of Th 4 f and Th 4 d bands after the adsorption of Th(IV) for both materials. The area of O 1 s peak at 531.0 eV increased significantly after adsorption, indicating that it was the hydrolyzed thorium ions that interacted with the N sites. The peak at 399.1 and 400.3 eV in the N 1 s spectrum of Py-TFImI-25 COF before Th(IV) adsorption (Fig. 4c) could be attributed to intramolecular imine-bonding ($N_{-C=N-}$), and the N atom bonding to the methyl group on the cationic benzimidazolium linkers ($N_{Im-CH_3}$). After the uptake of Th(IV), the new peak at 398.3 eV can be assigned to the Th-N coordination bond. In the N 1 s spectrum of Py-TFIm-25 COF before Th(IV) adsorption (Fig. 4d), the peak at 398.9, 399.4 and 400.4 eV could be attributed to imine-bonding ($N_{-C=N-}$), the N atom on benzimidazole linkers unbound to the methyl group ($N_{Im}$), and the N atom bonding to the methyl group ($N_{Im-CH_3}$). After the uptake of Th(IV), the new peak at 398.2 eV corresponding to the Th-N

coordination bond was observed. In addition, the energy dispersive spectrometer (EDS) further confirmed the adsorption of Th(IV) on Py-TFImI-25 COF and Py-TFIm-25 COF (Fig. 4e). These results suggested that the selective capture by Py-TFImI-25 COF and Py-TFIm-25 COF on Th(IV) was via Th-N coordination interaction.

To identify the plausible adsorption sites on Py-TFImI-25 COF and Py-TFIm-25 COF, the localized electronic distributions of the N sites were first investigated. Figure 5a, b showed the electronic localization function (ELF) contour maps of the Py-TFImI-25 COF skeleton focusing on the $N_{Im-CH_3}$ (Fig. 5a) and the $N_{-C=N-}$ (Fig. 5b). The higher ELF value observed at the $N_{-C=N-}$ indicated that the highly localized charge density, compared with the $N_{Im-CH_3}$ where the lone pair electrons of N atoms on the cationic benzimidazolium linkers have been coupled with the −CH₃ motif. Therefore, the $N_{-C=N-}$ can be a potential electron donor which is more advantageous to the adsorption[56,57]. Figure 5c, d showed the ELF contour maps of the Py-TFIm-25 COF skeleton focusing on the $N_{Im}$ (Fig. 5c) and $N_{-C=N-}$ (Fig. 5d). The higher ELF value observed at the $N_{Im}$ indicates that the $N_{Im}$ can be a potential electron

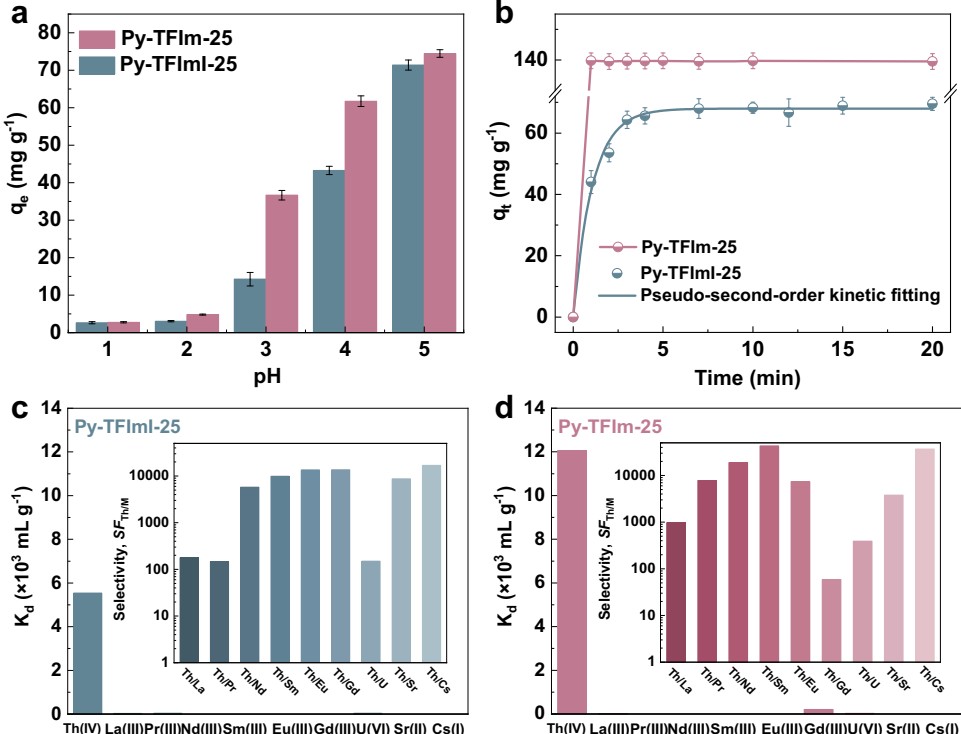

**Fig. 3 | Capture performance of Py-TFImI-25 COF and Py-TFIm-25 COF. a** Effect of pH on removal of Th(IV). **b** Adsorption kinetics. **c** and **d** The distribution coefficients of Th(IV) and competing ions. The inset shows the separation factors of Th(IV) and competing ions. All the error bars represent the standard deviation of the experiments.

donor to interact with the metal cations. The ELF values follow the order $N_{Im} > N_{-C=N-} > N_{Im-CH_3}$. Compared with the ELF of $N_{-C=N-}$ over Py-TFImI-25 COF (Fig. 5a), the $N_{Im}$ over Py-TFIm-25 COF (Fig. 5c) possesses a relatively higher ELF value, indicating that the $N_{Im}$ site of Py-TFIm-25 COF is a fairly stronger electron donor, which is consistent with the experimental result that Py-TFIm-25 COF has a higher Th(IV) adsorption capacity.

Then, the adsorption potential energies of metal cations on the rational N sites of Py-TFImI-25 COF and Py-TFIm-25 COF adsorbent were examined. Firstly, we illustrated the interaction mechanism of Th(IV) with $N_{-C=N-}$ sites of Py-TFImI-25 COF and calculated the adsorption potential energies. The precise aqueous speciation of Th(IV) has shown to be the function of pH[58,59]. Due to that the main form of thorium ion is $[Th(OH)_3(H_2O)_4]^+$ in the experimental pH range, so the adsorption process of $[Th(OH)_3(H_2O)_4]^+$ was investigated. As shown in Fig. 6a, the $[Th(OH)_3(H_2O)_4]^+$ episode interacts with the $N_{-C=N-}$ in Py-TFImI-25 COF skeleton, with the calculated Th-N bond distance of 2.586 Å. One $H_2O$ molecule is released during the adsorption process, due to the steric hindrance of the adsorption site (Supplementary Fig. 23). The calculated adsorption energy is -105.5 kcal mol$^{-1}$, which is comparable with the case of Th(IV) interacting with Py-TFImI-25 COF skeleton under a solvent environment (Figs. 6b, -106.9 kcal mol$^{-1}$). This indicated that the models of metal cations used in the subsequent parts are rational, which is also agreeable with previous studies[55,56].

The calculated adsorption potential energies between metal cations and the adsorbents were summarized in Fig. 6b. The adsorption configurations are the same for all different cations, see Supplementary Figs. 24 and 25. The $\Delta E_{ads}$ values for all adducts are negative indicating the $N_{-C=N-}$ on Py-TFImI-25 COF and the $N_{Im}$ on Py-TFIm-25 COF are the adsorption sites. It is worth noting that the absolute values of the adsorption energies on Py-TFIm-25 COF are all larger than the corresponding energies on Py-TFImI-25 COF. This indicated that the metal cations adsorption strength on the $N_{Im}$ of Py-TFIm-25 COF is

elevated to some extent, which is consistent with the ELF analysis above. For both of them, the adsorption strength of Th(IV) on COF is the strongest, and the remaining ions are poorly adsorbed, which is in agreement with the experimental results.

To acquire a thorough understanding of why the materials selectively capture Th(IV), the electronic density of state (DOS) and the charge density difference before and after metal cations adsorbed on the adsorbents were investigated. As shown in Fig. 7, the density of electrons on the $N_{-C=N-}$ of Py-TFImI-25 COF and $d$ and $f$ orbitals of Th(IV) significantly changed after the adsorption of Th(IV). This phenomenon illustrated that the electrons of $N_{-C=N-}$ transfer to the $d$ and $f$ orbitals of Th(IV) during Th(IV) adsorption. To figure out the separation of Th(IV)/ Ln(III), we used Nd(III) and Sm(III) to represent the Ln(III) and compared the change in the density of electrons of their $d$ and $f$ orbitals before and after adsorption and found that in contrast to the $5f$ orbitals of Th(IV), there is little electron transferred in the $4f$ orbitals of Nd(III) and Sm(III). Because the energy difference between $5f$ and $6d$ orbitals of actinides is smaller than that between $4f$ and $5d$ of lanthanides, so electrons are prone to jump between $5f$ and $6d$ orbitals. As a result, the $5f$ orbitals of actinides are more receptive to electrons than the $4f$ orbitals of lanthanides and can be more easily coordinated to N sites. The separation of Th(IV)/U(VI) is due to that the $d$ and $f$ orbitals of U(VI) are much less capable of receiving electrons than Th(IV). Since U(VI) primarily coordinates with oxygen ions to form stable uranyl ions, and thus the ability to coordinate with N is weakened. Meanwhile, almost no electrons transfer in Sr(II) and Cs(I) adsorption.

The charge density difference also illustrated that the degree of the amount of the transferred electrons gradually decreases with the binding strength between the metal cation with Py-TFImI-25 COF weakening, which is in accordance with the results of DOS. The surface of the single Th(IV) ion has a large amount of electronic charge accumulation. The bond length of Th-N is 2.790 Å, which is comparable with the simulated results of Ding et al.[55].

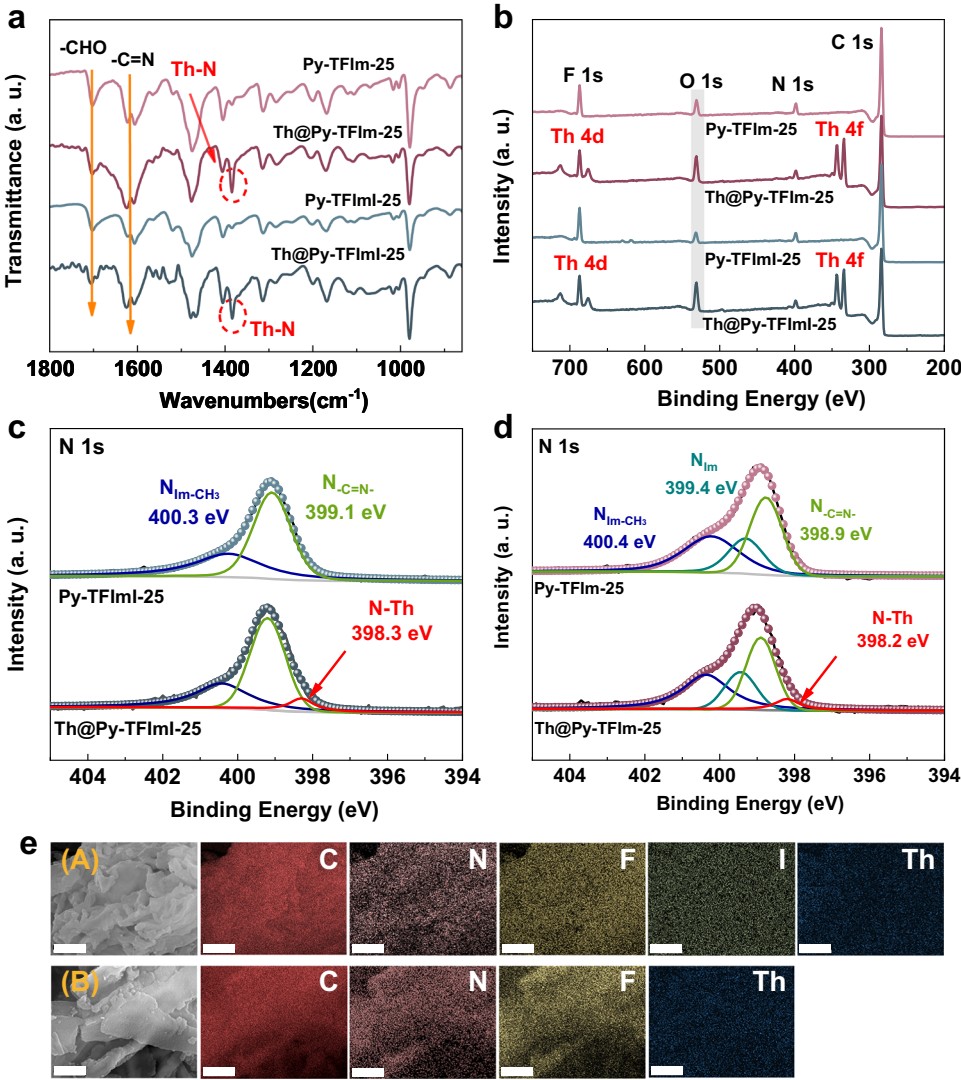

**Fig. 4 | Characterizations of Th(IV)/adsorbents interaction. a, b a** FT-IR and **b** XPS spectra of Py-TFImI-25 COF and Py-TFIm-25 COF before and after Th(IV) adsorption. **c** N 1s XPS spectra of Py-TFImI-25 COF before and after adsorption. **d** N 1s XPS spectra of Py-TFIm-25 COF before and after adsorption. **e** EDS mapping of Py-TFImI-25 COF (A) and Py-TFIm-25 COF (B) after adsorption. (Scale bar, 5 μm).

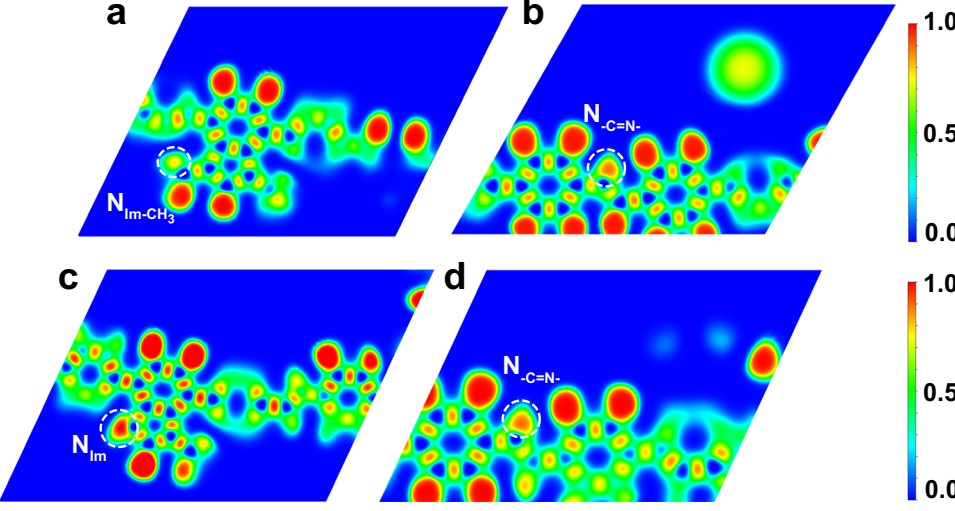

**Fig. 5 | DFT calculations.** ELF contours focusing on **a** the N atom of Im-CH₃, **b** the N atom of imine of Py-TFImI-25 COF, **c** the N atom of Im, and **d** the N atom of imine of Py-TFIm-25 COF.

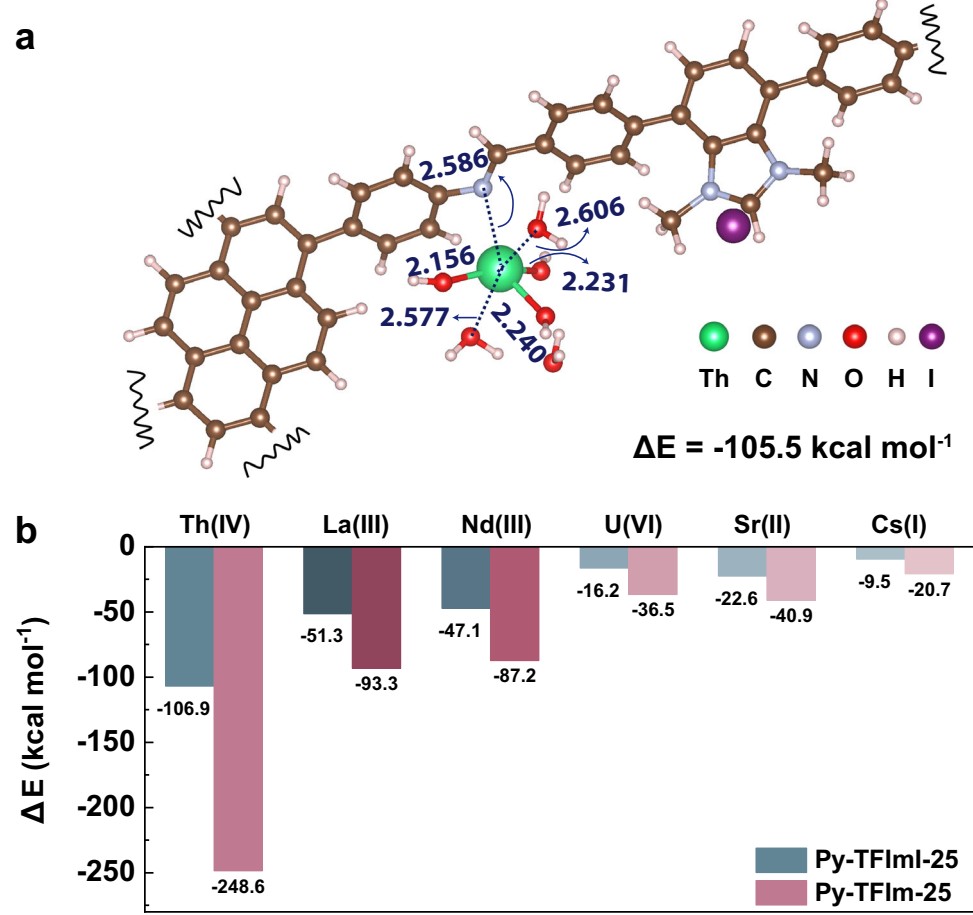

**Fig. 6 | DFT calculations. a** The optimized structures of the most energetically stable of [Th(OH)₃(H₂O)₃]⁺-Py-TFImI-25 COF. **b** The adsorption potential energies ($\Delta E_{ads}$) (kcal mol⁻¹) between metal cations and Py-TFImI-25 COF/Py-TFIm-25 COF adsorbents.

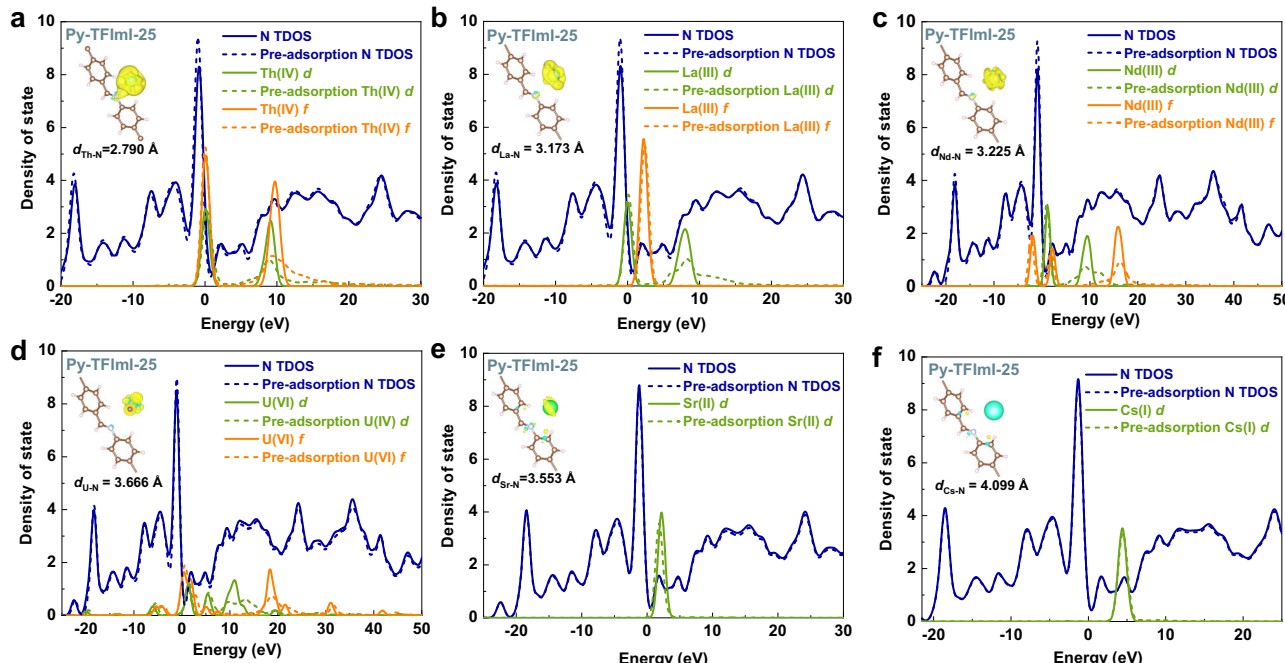

**Fig. 7 | DFT calculations of DOS and charge density difference of Py-TFImI-25 COF. a–f** DOS of various metal cations adsorption on Py-TFImI-25 COF. The inset shows the calculated isosurface (level 0.007) of charge density difference upon the adsorption of the mental cations on the N atom of imine.

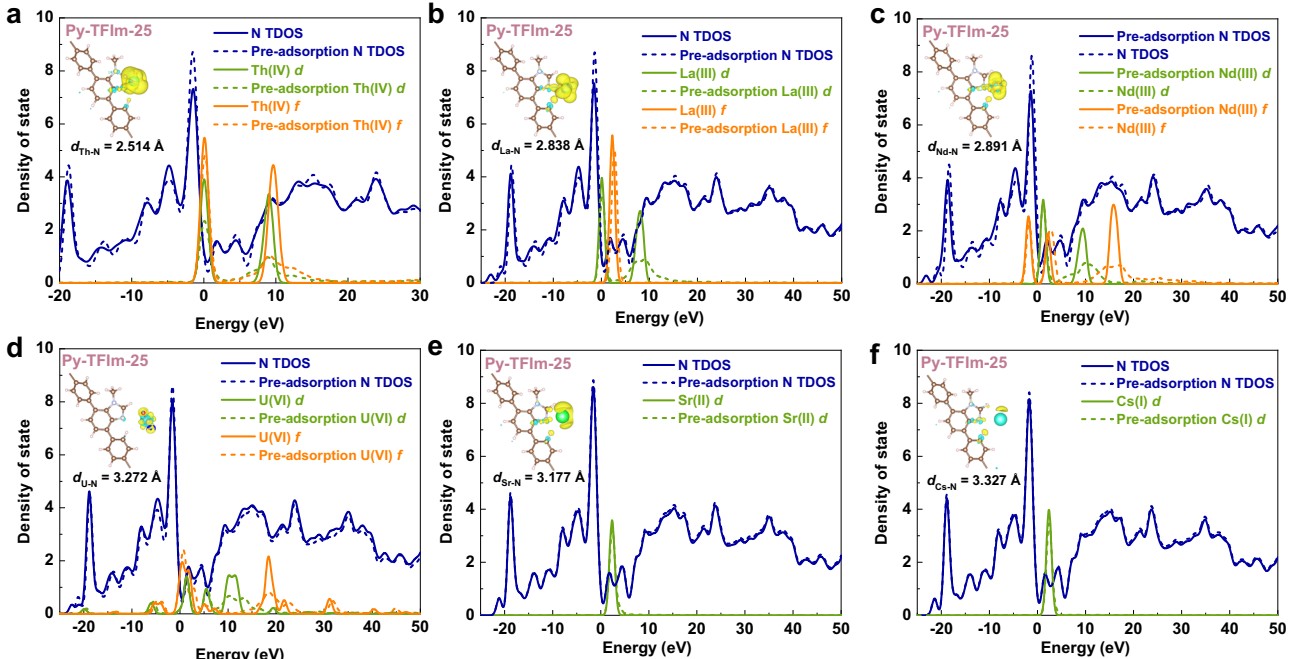

**Fig. 8 | DFT calculations of DOS and charge density difference of Py-TFIm-25 COF. a–f** DOS of various metal cations adsorption on Py-TFIm-25 COF. The inset shows the calculated isosurface (level 0.007) of charge density difference upon the adsorption of the mental cations on the N atom of Im.

The calculated DOS before and after Th(IV) adsorption on Py-TFIm-25 COF were demonstrated in Fig. 8. Take Th(IV)-Py-TFIm-25 COF as an example, compared with the DOS before adsorption, there is a distinct transfer of electrons from N atom to $d$ and $f$ orbitals of Th(IV). The $d$ and $f$ orbitals of Th(IV) interacting with the $N_{Im}$ site on Py-TFIm-25 COF are more electron receptive than the $d$ and $f$ orbitals of Th(IV) interacting with the $N_{-C=N-}$ site on Py-TFImI-25 COF. The extent of charge transfer faded following the same sequence of the adsorption strength of metal cations on Py-TFIm-25 COF, see the insets of Fig. 8. The mechanism of selective adsorption towards Th(IV) by Py-TFIm-25 COF is similar to that of Py-TFImI-25 COF. Those electronic analyses corroborated the obvious electron transfer between the metal cations and $N_{Im}$, which is consistent with the experimental and adsorption strength analysis.

In conclusion, the development of adsorbents with high adsorption selectivity and capacity relies on the understanding of the relationship between structure and performance. we speculated that adsorbents containing N sites may exhibit promising affinity and selectivity for thorium ions based on previous investigations, and synthesized an ionic COF named Py-TFImI-25 COF and its deionization analog named Py-TFIm-25 COF. The two COFs exhibited similar structure characteristics, being different only in N sites, allowing the investigation of the role of N sites on the capture of Th(IV). Py-TFIm-25 COF exhibited a significantly higher Th(IV) uptake capacity and adsorption rate than Py-TFImI-25 COF. The multi-component adsorption performance revealed that both of the two COFs have a high affinity for Th(IV), but hardly adsorb other elements, suggesting good selectivity towards Th(IV). The adsorption process can be illustrated as that one $H_2O$ molecule in the hydrolyzed thorium ion was replaced by the nitrogen atom in the COF structure, thereby forming the Th-N coordination bond. The ELF contour maps combined with calculated adsorption potential energies revealed that Th(IV) binding affinity at different N sites follows the order $N_{Im} > N_{-C=N-} > N_{Im-CH_3}$. The DOS and charge density difference suggested that the separation of Th(IV)/Ln(III) is due to the energy difference between $5f$ and $6d$ orbitals of actinides being smaller than that between $4f$ and $5d$ of lanthanides. The separation of Th(IV)/U(VI) is due to that U(VI)

mainly coordinates with oxygen ions to form stable uranyl ions, and thus the ability to coordinate with N is weakened. The above results indicated that $N_{Im}$ and $N_{-C=N-}$ as the door atoms both have highly selective receptors for Th(IV), especially for $N_{Im}$, which is a more effective site to not only improve adsorption capacity but also enhance selectivity. Through the above basic research, we hope to provide high-performance materials for the treatment of Th(IV)-containing radioactive waste streams by realizing the directed assembly of highly functionalized COFs through rational feedback on molecular structure and design.

## Data availability

All data generated in this study are provided in the article and the Supplementary Information or are available from the corresponding author upon request. Source data are provided with this paper.

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

## Acknowledgements
The authors gratefully acknowledge the supports from the Continuous-Support Basic Scientific Research Project (BJ22003103 to Y.O.), and the National Natural Science Foundation of China (No. 21790371 to G.Y.).

## Author contributions
G.Y., Y.O., S.X., and J.Z. conceived and designed the research. X.L. and J.Z. designed the experiments. M.W., Y.G., X.L., and J.Z. prepared and characterized the materials. X.L., F.G., and T.J conducted the adsorption experiments. X.L., K.M., and H.S. performed PXRD simulations. X.L. and W.X. performed the DFT calculations. X.L., F.G., and J.Z. wrote the manuscript with contributions from all the authors.

## Competing interests
The authors declare no competing interests.
