## [Peer Review File · Nature Communications]

Efficient and selective capture of thorium ions by a covalent organic frameworkReviewers' Comments:

Reviewer #1:

Remarks to the Author:

In this manuscript, the authors done an excellent research work in which they demonstrated the synthesis and structural characterization (Pawley refinement using PXRD data) of a two new mesoporous COFs namely Py-TFImI-25 COF and Py-TFIm-25 COF. The as-synthesized COFs are capable for coordination bonding with the Th(IV) through the N atoms of the COFs resulted in an outstanding selectivity and high Th(IV) adsorption capacity. Py-TFIm-25 COF exhibited a significantly higher Th(IV) uptake capacity and adsorption rate than Py-TFImI-25 COF, which also outperformed the majority of previously reported adsorbents. Overall, the results presented in this manuscript are of potential to be published in "Nature Communications" citing major revisions as detailed below.

1. Authors did not discuss the challenges behind the selective separation of thorium from rare earth elements or uranium.
2. CIF files and checkcif reports should be provided if applicable. Also, details on methodology for structure simulation should be provided. Space group P1 sounds suspicion for such symmetric structure.
3. NMR – supporting information: Assign the peaks of the NMR with the corresponding protons having equivalent chemical environment.
4. How stable is the mesoporous COFs in water? PXRD and BET data should be provided after soaking the COF crystals in water.
5. It is not clear if as-syn or activated samples were used for this adsorption study. Since the COF pores are occupied with solvent molecules, is there any role of lattice solvent for the adsorption of Th(IV)? The TGA data shows presence of solvent molecules in the crystal lattice.
6. Is there is any role of particle size (exfoliation of 2D COF sheets) for the adsorption capacity of Th(IV)?
7. The experimental condition for the competitive adsorption of Th(IV) from the competing cations are not included in the manuscript?
8. The crystal structure stability of the COFs should be confirmed by performing PXRD of samples after Th(IV) adsorption.
9. Page 18, line 298: "The bond length of Th-N is 2.801 Å, which is comparable with the simulated results of Xia et. al." How did the authors determined the Th-N bond length? Computationally? The standard deviation (esd) is missing.
10. The Th(IV)-N-coordination in Py-TFImI-25 COF is not well described. Is it through imidazole N atom or -N-C=N-? The authors claim that the Th(IV) binding affinity at different N sites follows the order NIm > N-C=N- > NIm-CH3.
11. The authors mentioned that they performed ICP-MS for the quantification of Th(IV) and other metal ions (in supporting information), however, the quantified amount of metals are not discussed in the manuscript.
12. The experimental details for the activation (degassing) of the COF samples for the BET experiment are missing.

Reviewer #2:

Remarks to the Author:

In this manuscript, Liu et al. describes their synthesis of imine COFs featuring imidazole and imidazolium sub-structures and demonstrates that the former exhibits greater Thorium capture power. From that, the authors hypothesize the role of coordinating nitrogen in the capture mechanism. In our opinion, this is an interesting piece of experimental information. However, it is not surprising as the effect of nitrogen sites in metal ion sorption has been well known. Furthermore, we find the study lacked of important data and control experiments to support the claimed theory. Therefore, we would not recommend this manuscript to be published in Nature Communications. Below are our detail comments and questions:

1. The authors state that based on their calculation the Th(IV) affinity follows the order NIm > N-C=N- > NIm-CH₃. That means imidazole nitrogens bind to Th(IV) more strongly than imine nitrogens. Interestingly, their own previously reported 3D COF, namely COF-DL229, (<https://doi.org/10.1016/j.seppur.2022.121413>) possessed exclusively imine linkages and exhibited much higher Th(IV) uptake capacity (~500 mg/g vs ~ 140 mg/g of Py-TFIm-25 of this study). We have to mention that the COF-DL229 has a lower surface area (569 m²/g vs 1430 m²/g of Py-TFIm-25) and a narrower pore size (1.68 nm vs 2.3 nm). That indicates that Th(IV) can penetrate through Py-TFIm-25 more easily (because of larger pores) and be more exposed to nitrogen sites (because of higher surface area). Yet, much lower capacity was obtained with Py-TFIm-25. Does it disprove the claimed higher affinity of imidazole nitrogen mentioned above?
2. The role of the fluorinated fragment from 2',3',5',6'-tetrafluoro-[1,1':4',1''-terphenyl]-4,4''-dialdehyde (TFTDA) is unclear. Why do the authors have to use this particular building block?
3. The XPS spectra show that N-Th is observed from the use of Py-TFIm-25. This means the imine nitrogen is the actual binding sites because the all of the nitrogens in the imidazolium have no free electrons to coordinate. From Fig. 3A, we can see that at pH 5 where the nitrogens are most neutral, Py-TFImI-25 and Py-TFIm-25 perform nearly equivalent. That appears to us that the imine nitrogens play a significant role in the capturing power. At too low pHs (pH 1 and 2), the imine bond is more susceptible to hydrolysis and therefore the same phenomenon is observed. At pH3 and 4, bigger capacity differences are recorded due to the fact that the imines become protonated and hence exist as cations that exert electrostatic repulsion to Th(IV) and this phenomenon is more dramatic in the Py-TFIm-25.
4. The water stability of the COFs are not assessed. It is known that imine is not stable in water as it can be easily hydrolyzed in both acidic and basic conditions.
5. A linear relationship between imidazole content and uptake capacity should be observed if the authors statement about its dominating binding strength. Therefore, a series of COFs containing 1,2,3 and all 4 imidazole units should be obtained and evaluated.
6. Structures in Fig. 1 need revising: TFTDA in Fig. 1b; pyrene moiety sometimes lack complete conjugation.

Reviewer #3:

Remarks to the Author:

The proposed manuscript presents a detailed study on the effect of different N sites on Th(IV) uptake by designing two analog COF materials and the electron-scale selective adsorption mechanism. The authors clearly described the structure and adsorption characterization of the obtained materials, as well as the in-depth investigation of the adsorption behavior. The study is well designed and the conclusion can be supported by the experimental data presented. So, this manuscript is recommended for publication in Nature Communications after the following issues are addressed.

1. There are structural formula errors in Fig.1. Please note the correct drawing of the pyrene unit.
2. "The removal of I- from Py-TFImI-25 COF modifies the interlayer π -electron environment" in Line 166 is not rigorous. Since from ELF contour maps, the localized electronic distribution of N-C=N- over Py-TFImI-25 COF is similar to N-C=N- over Py-TFIm-25 COF.
3. It is recommended to increase the Th(IV) concentration or reduce the amount of Py-TFIm-25 COF to catch the rising phase of the kinetic curve.
4. The optimized structures of [Th(OH)₃(H₂O)₄]⁺ and Py-TFImI-25 COF should be supplemented so as to reflect the changes during the adsorption process.
5. It is puzzling that the calculated adsorption potential energy of Py-TFIm-25 COF on La(III) is comparable to that of Py-TFImI-25 COF on Th(IV), but the selective adsorption experiments show that Py-TFIm-25 COF hardly adsorbs La(III).
6. The source of the main materials should be marked.

Response to reviewer #1:

Thank you for your useful comments on improving the quality of this manuscript. We have revised the manuscript according to your comments. The revised parts are highlighted in red in the revised manuscript.

The point-to-point responses are summarized below:

Comment 1: Authors did not discuss the challenges behind the selective separation of thorium from rare earth elements or uranium.

Response: Thank you for your careful review and for pointing this out. The separation of thorium from rare earth elements and uranium is quite challenging, because of their similar chemical properties (*Nat. Commun.* **2023**, *14*, 261; *Nanoscale.* **2020**, *12*, 1339-1348).

Correction:

However, the separation of thorium from rare earth elements and uranium is quite challenging, because of their similar chemical properties^{11,12}.

(Line 41-43, Page 3)

Comment 2: CIF files and check if reports should be provided if applicable. Also, details on methodology for structure simulation should be provided. Space group P1 sounds suspicious for such symmetric structure.

Response: Thank you for your valuable and thoughtful comment. We have added the CIF files for studying the selective adsorption mechanism in the supplementary information of the revised manuscript. The Material Studio software was used to construct the Py-TFImI-25 and Py-TFIm-25 COF and was not analyzed by the X-ray single crystal data, hence there is no checkcif report. The details on methodology for structural DFT-simulation are as follows and also added in the supplementary information of the revised manuscript.

In this work, the crystal models of Py-TFImI-25 and Py-TFIm-25 COF were constructed with the Crystal Building module of the Material Studio software. Then, a series of geometry optimizations were performed with UFF force field at the ultra-fine quality in the Forcite module, allowing optimization of all lattice parameters and atomic

coordinate. All the optimized COF materials were used for subsequent simulation of the powder diffraction patterns with the Reflex module in the Material Studio software. Both the eclipsed AA and staggered AB stacking modes were tried for the Py-TFImI-25 and Py-TFIm-25 synthesized here. Compared the experimental XRD results with the simulated XRD, finally, the eclipsed AA stacking modes were settled due to the good match of the experimental XRD results with the simulated XRD on the eclipsed AA stacking mode. Based on the obtained results, periodic density functional theory (DFT) calculations were performed with the cell parameters of $39.2073 \times 12.5936 \times 51.7061 \text{ \AA}^3$, $\alpha = \beta = \gamma = 90.000^\circ$ and $41.8840 \times 11.6685 \times 49.6189 \text{ \AA}^3$, $\alpha = \beta = \gamma = 90.000^\circ$ for Py-TFImI-25 and Py-TFIm-25 COF, respectively. During the structural DFT optimization, all the atoms in the periodic Py-TFImI-25 COF, Py-TFIm-25 COF, and the adsorbed heavy metal ions were allowed to be fully relaxed. The Pawley refinement was performed to iteratively optimize the lattice parameters until the Rwp value converges and the observed profile fitted well with the refined one. Detailed results are in Supplementary Table 1 and 2. (Page 16 in Revised supplementary information)

The structure building expressed that both materials are P1 space group. We suppose that it is because the linking units are composed of 25% BFIm (or BFIm) and 75% TFTDA, thus leading to such incomplete symmetry.

Comment 3: NMR – supporting information: Assign the peaks of the NMR with the corresponding protons having equivalent chemical environment.

Response: Thank you for the thoughtful comment. The peaks of ^1H NMR spectra of all compounds we synthesized were assigned and the ^1H NMR spectra were supplemented in the revised supplementary information.

Comment 4: How stable is the mesoporous COFs in water? PXRD and BET data should be provided after soaking the COF crystals in water.

Response: Thank you for your professional comments. We have complemented relevant experiments and demonstrated that both materials exhibited good stability after

soaking in water and maintained the integrity of the skeleton and the crystal structure (see Fig. R1 and R2).

Fig. R1 PXRD patterns of Py-TFImI-25 COF and Py-TFIm-25 COF after soaking in water. (Solid-liquid ratio = 1:3000 g mL⁻¹, Shaking time = 1 h, T= 25°C).

Fig. R2 N₂ sorption isotherms of Py-TFImI-25 COF and Py-TFIm-25 COF after soaking in water. (Solid-liquid ratio = 1:3000 g mL⁻¹, Shaking time = 1 h, T= 25°C).

To provide more details, we added some sentences in the revised manuscript, as follows:

Correction:

The two materials exhibited good stability after soaking in water and maintained the integrity of the skeleton and the crystal structure (Supplementary Fig. 10 and 11).

(Line 148-150 Page 8)

Comment 5: It is not clear if as-syn or activated samples were used for this adsorption study. Since the COF pores are occupied with solvent molecules, is there any role of lattice solvent for the adsorption of Th(IV)? The TGA data shows presence of solvent molecules in the crystal lattice.

Response: Thank you for the kind comments. It is the as-synthesized samples that were used for the adsorption experiments. During the synthesis of the materials, the solvent molecules were removed as much as possible during repeated washing with THF and acetone, and the detergent was removed by vacuum drying at 120 °C. As shown in Fig. R3, the DTG curves showed that the materials have only one obvious weight loss peak at 535 °C, which was attributed to the sustained thermal decomposition of the skeleton. The BET surface areas of the synthesized materials are about 1400 m² g⁻¹, which is considerably high with the ionic COF reported (*Angew. Chem. Int. Ed.* **2016**, *55*, 1737-1741; *J. Am. Chem. Soc.* **2016**, *138*, 5897-5903; *Angew. Chem. Int. Ed.* **2017**, *56*, 4982-4986). Therefore, we believe that the solvent molecules have been removed as much as possible during the synthesis process, and the weight loss of less than 10% of the material in the 25-500°C range is due to the adsorption of water from the air because of the large specific surface area. Therefore, no further activation was performed, which is also consistent with other COF treatments used for the adsorption of metal ions in the literature (*Angew. Chem. Int. Ed.* **2020**, *59*, 4168-4175; *CCS Chemistry.* **2019**, *1*, 286-295).

Fig. R3 TG-DTG curves of Py-TFImI-25 COF and Py-TFIm-25 COF.

Comment 6: Is there is any role of particle size (exfoliation of 2D COF sheets) for the adsorption capacity of Th(IV)?

Response: Thank you for your professional question. We supposed that particle size (exfoliation of 2D COF sheets) can play a significant role in the adsorption capacity

of Th(IV) by influencing the surface area. The exfoliation of 2D COF sheets can lead to a larger surface area, which can increase the number of effective adsorption sites, thus improving the Th(IV) adsorption capacity. Since Py-TFImI-25 COF and Py-TFIm-25 COF were synthesized with only the linking units BFIIIm and BFIm different, all other synthesis conditions were parallel. So, the specific surface areas of the materials are similar ($1324.05 \text{ m}^2 \text{ g}^{-1}$ for Py-TFImI-25 COF, and $1430.15 \text{ m}^2 \text{ g}^{-1}$ for Py-TFIm-25 COF), which allows us to explore the effect of the different N sites caused by BFIIIm and BFIm on the adsorption.

Comment 7: The experimental condition for the competitive adsorption of Th(IV) from the competing cations are not included in the manuscript?

Response: Thank you for your valuable comment. The experimental condition for the competitive adsorption of Th(IV) from the competing cations is in Section 1.3.4 in the supplementary information. Also, we have added some details to the description of the experimental conditions, as follows.

Correction:

1.3.4 Adsorption selectivity

A multi-ion solution of Th(IV), U(VI), Sr(II), Cs(I), La(III), Pr(III), Nd(III), Sm(III), Eu(III), and Gd(III) was prepared with each metal ion maintaining at a concentration of around 25 mg L^{-1} and the pH was adjusted to 4. The solid-liquid ratio was 1:3000 g mL^{-1} by adding 1.5 mg COF materials into 4.5 mL multi-ion solution. The batch adsorption experiments were carried out at a temperature of $25 \text{ }^\circ\text{C}$ and a speed of 200 rpm, and then the multi-ion solution after adsorption was separated with a $0.22 \text{ }\mu\text{m}$ aqueous nylon filter to measure the concentration of Th(IV) and the competing ions.

(Page 16 in Revised supplementary information)

Comment 8: The crystal structure stability of the COFs should be confirmed by performing PXRD of samples after Th(IV) adsorption.

Response: Thank you for your valuable suggestion. We supplemented the PXRD characterization of the two materials after Th(IV) adsorption. PXRD patterns of Th(IV)-

loaded samples were carried out on Rigaku Ultima IV, since thorium is a radioactive element. As illustrated in Fig. R4, the obtained PXRD patterns of Py-TFImI-25 COF and Py-TFIm-25 COF after Th(IV) adsorption exhibited consistent diffraction peaks with the pristine crystalline structures, indicating that the Th-N coordination would not disrupt the crystalline structure.

Fig. R4 PXRD patterns of Py-TFImI-25 COF and Py-TFIm-25 COF before and after Th(IV) adsorption.

We have added the relevant PXRD results in the revised manuscript, as follows:

Correction:

Various characterization approaches were adopted to investigate the interaction mechanism of Py-TFImI-25 COF and Py-TFIm-25 COF towards Th(IV). The obtained PXRD patterns of Py-TFImI-25 COF and Py-TFIm-25 COF after Th(IV) adsorption exhibited consistent diffraction peaks with the pristine ones (Supplementary Fig. 16), indicating that the adsorption of Th(IV) would not disrupt the crystalline structures.

(Line 210-213, Page 12)

PXRD patterns of Th(IV)-loaded samples were carried out on Rigaku Ultima IV.

(Page 3 in Revised supplementary information)

Comment 9: Page 18, line 298: “The bond length of Th-N is 2.801 Å, which is comparable with the simulated results of Xia et. al.” How did the authors determined the Th-N bond length? Computationally? The standard deviation (esd) is missing.

Response: Thank you for your valuable comment. The bond length of Th-N is

calculated computationally using density functional theory (DFT), as implemented in the CP2K code of the QUICKSTEP program by employing a mixed Gaussian and plane-wave basis sets. Since the bond length of Th-N is not obtained experimentally, there is no relevant standard deviation.

Comment 10: The Th(IV)-N-coordination in Py-TFImI-25 COF is not well described. Is it through imidazole N atom or N-C=N-? The authors claim that the Th(IV) binding affinity at different N sites follows the order $N_{Im} > N_{C=N} > N_{Im-CH_3}$.

Response: Thank you for your kind comment. The Th(IV)-N coordination in Py-TFImI-25 COF is through N-C=N-, as shown in Fig. 5 and 6. Since the lone pair electrons of imidazole N atom in Py-TFImI-25 COF have been coupled with the -CH₃ motif (Fig. 1a), so only the N-C=N- in Py-TFImI-25 COF could be the electron donor atom. To avoid such confusion, the relevant description was supplemented as follows.

Correction:

As shown in Fig. 6a, the $[Th(OH)_3(H_2O)_4]^+$ episode interacts with the N-C=N- in Py-TFImI-25 COF skeleton, with the calculated Th-N bond distance of 2.586 Å.

(Line 264-266, Page 16)

Comment 11: The authors mentioned that they performed ICP-MS for the quantification of Th(IV) and other metal ions (in supporting information), however, the quantified amount of metals are not discussed in the manuscript.

Response: Thank you for your careful review. It is neglected because the distribution coefficients of the two materials for competing ions are too small. We have added the following discussion in the manuscript as follows.

Correction:

Py-TFImI-25 COF exhibited a distribution coefficient of 5.54×10^3 mL g⁻¹ for Th(IV), a distribution coefficient ranging from 0.41 to 37.70 mL g⁻¹ for lanthanide ions, 37.04 mL g⁻¹ for U(VI), 0.64 mL g⁻¹ for Sr(II), and 0.33 mL g⁻¹ for Cs(I). Meanwhile, Py-TFImI-25 COF exhibited a distribution coefficient of 1.21×10^4 mL g⁻¹ for Th(IV), a

distribution coefficient ranging from 0.28 to 204.35 mL g⁻¹ for lanthanide ions, 30.80 mL g⁻¹ for U(VI), 3.18 mL g⁻¹ for Sr(II), and 0.33 mL g⁻¹ for Cs(I).

(Line 191-197, Page 11)

Comment 12: The experimental details for the activation (degassing) of the COF samples for the BET experiment are missing.

Response: Thank you for your careful review. We have supplemented the activation of the COF samples for the BET experiment as follows.

Correction:

The fresh sample was activated in the degassing station of the instrument at 30 °C for 2 h, and then at 120 °C for 13 h to make the pores guest-free.

(Page 4 in Revised supplementary information)

Special thanks to you for your careful review and constructive suggestions.

Response to reviewer #2:

Thank you very much for your valuable comments. We have added control experiments in response to the comments, revised the manuscript and marked all changes in red on the revised manuscript.

The point-to-point responses are summarized below:

Comment 1: The authors state that based on their calculation the Th(IV) affinity follows the order $N_{Im} > N_{C=N} > N_{Im-CH_3}$. That means imidazole nitrogens bind to Th(IV) more strongly than imine nitrogens. Interestingly, their own previously reported 3D COF, namely COF-DL229, (<https://doi.org/10.1016/j.seppur.2022.121413>) possessed exclusively imine linkages and exhibited much higher Th(IV) uptake capacity (~500 mg/g vs ~ 140 mg/g of Py-TFIm-25 of this study). We have to mention that the COF-DL229 has a lower surface area (569 m²/g vs 1430 m²/g of Py-TFIm-25) and a narrower pore size (1.68 nm vs 2.3 nm). That indicates that Th(IV) can penetrate through Py-TFIm-25 more easily (because of larger pores) and be more exposed to nitrogen sites (because of higher surface area). Yet, much lower capacity was obtained with Py-TFIm-25. Does it disprove the claimed higher affinity of imidazole nitrogen mentioned above?

Response: Thank you for your careful review. Since the adsorption capacity is related to the initial concentration of the solution. When the initial concentration of the Th(IV) solution was 300.2 mg L⁻¹, the Th(IV) uptake capacity of COF-DL229 was 508.1 mg g⁻¹. When the Th(IV) initial concentration was 274.5 mg L⁻¹, the adsorption capacity of Py-TFIm-25 COF could reach 822.6 mg g⁻¹ (Supplementary Fig. 13). With similar Th(IV) initial concentrations, Py-TFIm-25 COF containing imidazole nitrogens exhibited a higher adsorption capacity than COF-DL229 containing imine nitrogens. Combined with the adsorption behavior of its analog Py-TFIm-25 COF, we suggested that imidazole nitrogen possesses a higher affinity, which is also consistent with the theoretical calculations.

Comment 2: The role of the fluorinated fragment from 2',3',5',6'-tetrafluoro-[1,1':4',1''-terphenyl]-4,4''-dialdehyde (TFTDA) is unclear. Why do the authors have

to use this particular building block?

Response: Thank you for your professional question. The introduction of the fluorinated fragment into the building block (TFTDA) is to enhance the molecular interlayer interaction between the fluorinated build block and non-fluorinated build block (*J. Am. Chem. Soc.* **2013**, *135*, 546-549), which can improve the crystallinity and BET surface area of the COFs. Meantime, the chemical stability of the COFs can also be enhanced.

The role of the fluorinated fragment from TFTDA was added in the revised manuscript, as follows:

Correction:

The introduction of TFTDA is to enhance the molecular interlayer interaction between the fluorinated build block and non-fluorinated build block⁴⁸, which can improve the crystallinity and chemical stability of the COFs.

(Line 87-90, Page 5)

Comment 3: The XPS spectra show that N-Th is observed from the use of Py-TFImI-25. This means the imine nitrogen is the actual binding sites because all of the nitrogens in the imidazolium have no free electrons to coordinate. From Fig. 3A, we can see that at pH 5 where the nitrogens are most neutral, Py-TFImI-25 and Py-TFIm-25 perform nearly equivalent. That appears to us that the imine nitrogens play a significant role in the capturing power. At too low pHs (pH 1 and 2), the imine bond is more susceptible to hydrolysis and therefore the same phenomenon is observed. At pH3 and 4, bigger capacity differences are recorded due to the fact that the imines become protonated and hence exist as cations that exert electrostatic repulsion to Th(IV) and this phenomenon is more dramatic in the Py-TFImin-25.

Response: Thank you for your careful review and for pointing this out. With regard to this concern, we would like to interpret from three points. (1) we do agree that imine nitrogen is the actual binding site in Py-TFImI-25 COF, since the nitrogens in the imidazolium have no free electrons to coordinate. However, as shown in Fig. 1b and Fig. 5c, the nitrogens in benzimidazolium linkers unbound to the methyl group in Py-

TFIm-25 COF possess a higher localized charge density, which could be the potential electron donor. (2) Since nitrogens are most neutral at pH 5, as shown in Table R1, the Th(IV) removal rates of Py-TFImI-25 COF and Py-TFIm-25 COF are as high as 99.8% and 99.9%, respectively, and the concentration of Th(IV) after adsorption is close to the detection limit, so the initial concentration of $\sim 25 \text{ mg L}^{-1}$ at pH 5 is not sufficient to reflect the difference in the capture of the two materials, which is also the reason why pH 4 was chosen for subsequent experiments. However, as the initial concentration of Th(IV) continued to increase, the difference in the adsorption capacity caused by different N sites of the two materials became apparent (Supplementary Fig. 13). (3) As interpreted comment 4, both two materials exhibited good stability after soaking in water. At low pH (pH 1 and 2), the protonation of the nitrogen sites is the reason for the decreased adsorption capacity. At pH 3 and 4, more nitrogen sites become available, resulting in an increased adsorption capacity. The capacity difference is due to the different nitrogen adsorption sites of the two materials ($\text{N}_{\text{C=N}}$ of Py-TFImI-25 COF vs N_{Im} of Py-TFIm-25 COF).

Table R1 Th (IV) adsorption performances over Py-TFImI-25 COF and Py-TFIm-25 COF at pH 5.

Adsorbents	C_0 (mg L^{-1})	C_e (mg L^{-1})	q_e (mg g^{-1})	Removal rate (%)
Py-TFImI-25	24.34	0.05	72.87	99.8
Py-TFIm-25	25.23	0.03	75.60	99.9

Comment 4: The water stability of the COFs are not assessed. It is known that imine is not stable in water as it can be easily hydrolyzed in both acidic and basic conditions.

Response: Thank you for your suggestion. Although imine is not stable in water, COFs, which are entirely composed of organic units connected by covalent bonds, are known to have high chemical and thermal stability (*ChSRv.* **2013**, *42*, 548-568). As shown in Fig. R1 and R2, both two materials exhibited good stability after soaking in water and maintained the integrity of the skeleton and the crystalline structure. In addition, in our subsequent study of the fluorescence property of Py-TFImI-25 COF, as shown in Fig. R3, Py-TFImI-25 COF have not been hydrolyzed in acid solutions,

otherwise, the fluorescence intensity would not be enhanced with increasing acidity.

Fig. R1 PXRD patterns of Py-TFImI-25 COF and Py-TFIm-25 COF after soaking in water. (Solid-liquid ratio = 1:3000 g mL⁻¹, Shaking time = 1 h, T= 25°C).

Fig. R2 N₂ sorption isotherms of Py-TFImI-25 COF and Py-TFIm-25 COF after soaking in water. (Solid-liquid ratio = 1:3000 g mL⁻¹, Shaking time = 1 h, T= 25°C).

Fig. R3 **a** Fluorescence spectra of Py-TFImI-25 COF in various acidic conditions with pH values from 0.02 to 2.98 ($\lambda_{\text{ex}} = 360 \text{ nm}$). **b** Eye naked color (above) and fluorescence image (below) changes of Py-TFImI-25 COF after immersion in varieties of acidic solutions from pH 1 to pH 6 (from left to right).

We have supplemented the water stability of the two COFs in the revised manuscript, as follows:

Correction:

The two materials exhibited good stability after soaking in water and maintained the integrity of the skeleton and the crystal structure (Supplementary Fig. 10 and 11).

(Line 148-150, Page 8)

Comment 5: A linear relationship between imidazole content and uptake capacity should be observed if the authors statement about its dominating binding strength. Therefore, a series of COFs containing 1,2,3 and all 4 imidazole units should be obtained and evaluated.

Response: Thank you for your valuable suggestion. We choose to use 1 imidazole unit and 3 TFTDA units as linkers instead of using 4 imidazole units due to that the introduction of TFTDA can improve the crystallinity and chemical stability of the COFs, as interpreted in comment 2. Owing to the complexity of synthesis, it has been almost a year since the start of synthesis. Considering the research cycle of the subject, the aim at this stage is to verify the possibility of the application of the two COFs in this field and figure out the adsorption mechanism. Based on the present results proving that imidazole nitrogens are more effective adsorption sites, as suggested, we would like to continue synthesizing COFs containing 1,2,3 and all 4 imidazole units and find the linear relationship between imidazole content and uptake capacity to develop the maximum potential of this class of the materials.

Comment 6: Structures in Fig. 1 need revising: TFTDA in Fig. 1b; pyrene moiety sometimes lack complete conjugation.

Response: Thank you for pointing this out. We have corrected them in the revised manuscript, as follows.

Correction:

Fig. 1 Schematic illustration of the synthesis of **a** Py-TFImI-25 COF and **b** Py-TFIm-25 COF.

(Line 104-105, Page 6)

Once again, thank you very much for your comments and suggestions.

Response to reviewer #3:

Thank you very much for your important and helpful comments. We have revised the manuscript following the comments and marked all the amends in red on our revised manuscript.

The revision details are listed as follows:

Comment 1. There are structural formula errors in Fig.1. Please note the correct drawing of the pyrene unit.

Response: Thank you for your careful review. We have corrected them in the revised manuscript, as follows.

Correction:

Fig. 1 Schematic illustration of the synthesis of a Py-TFImI-25 COF and b Py-TFIm-25 COF.

(Line 104-105, Page 6)

Comment 2: "The removal of I- from Py-TFImI-25 COF modifies the interlayer π -electron environment" in Line 166 is not rigorous. Since from ELF contour maps, the localized electronic distribution of N-C=N- over Py-TFImI-25 COF is similar to N-C=N- over Py-TFIm-25 COF.

Response: Thank you for your professional recommendation. The localized electronic distribution of N-C=N- over Py-TFImI-25 COF is indeed similar to N-C=N- over Py-TFIm-25 COF. Meanwhile, we calculated the Bader charge numbers of N sites on Py-TFIm-25 COF and Py-TFImI-25 COF, and the results showed that N-C=N- on Py-TFImI-25 COF and N-C=N- on Py-TFIm-25 COF have similar electronegativity (Table R1).

Table R1 Bader charge number (unit: $|e|$) of N sites over Py-TFImI-25 COF and Py-TFImI-25 COF at pH 5.

Configuration	N _{Im} /N _{Im-CH₃}	N-C=N-
Py-TFImI-25	-1.821	-1.979
Py-TFIm-25	-2.165	-1.983

So, we have removed the relevant description and made the following revisions in the revised manuscript.

Correction:

The results showed that although Py-TFImI-25 COF and Py-TFIm-25 COF have analogous structures, the new N sites on the benzimidazolium linkers of Py-TFIm-25 COF could be responsible for the improved sorption performance.

(Line 170-173, Page 10)

Comment 3: It is recommended to increase the Th(IV) concentration or reduce the amount of Py-TFIm-25 COF to catch the rising phase of the kinetic curve.

Response: Thank you for your valuable recommendation. The objective of this subject is to extract trace concentrations of thorium ions from rare-earth raffinate. The concentration of thorium ions chosen for the kinetic experiments is close to that in the actual process. Due to the protonation of the material, the solid-to-liquid ratios of the kinetic experiments, thermodynamic experiments, and selectivity experiments were kept the same as much as possible, to eliminate the influence of the protonation on the

adsorption.

Comment 4: The optimized structures of $[\text{Th}(\text{OH})_3(\text{H}_2\text{O})_4]^+$ and Py-TFImI-25 COF should be supplemented so as to reflect the changes during the adsorption process.

Response: Thank you for your valuable recommendation. We have supplemented the optimized structures of $[\text{Th}(\text{OH})_3(\text{H}_2\text{O})_4]^+$ and Py-TFImI-25 COF in the supplementary information of the revised manuscript.

Correction:

As shown in Fig. 6a, the $[\text{Th}(\text{OH})_3(\text{H}_2\text{O})_4]^+$ episode interacts with the N_{C=N}- in Py-TFImI-25 COF skeleton, with the calculated Th-N bond distance of 2.586 Å. One H₂O molecule is released during the adsorption process, due to the steric hindrance of the adsorption site (Supplementary Fig. 17).

Supplementary Fig. 17 The optimized structures of the most energetically stable of **a** $[\text{Th}(\text{OH})_3(\text{H}_2\text{O})_4]^+$ and **b** Py-TFImI-25 framework.

(Line 266-267, Page 16)

Comment 5: It is puzzling that the calculated adsorption potential energy of Py-TFIm-25 COF on La(III) is comparable to that of Py-TFImI-25 COF on Th(IV), but the selective adsorption experiments show that Py-TFIm-25 COF hardly adsorbs La(III).

Response: Thank you for your professional question. For the Py-TFIm-25 COF adsorbent, the N_{Im} site is the strongest adsorption site for the heavy metal ions, while the N_{C=N}- is the adsorption site over Py-TFImI-25 COF. The calculated adsorption potential energy between Th(IV) and Py-TFIm-25 COF is $-235.5 \text{ kcal}\cdot\text{mol}^{-1}$, which is much lower than that of La(III) interacting with Py-TFIm-25 COF skeleton ($-90.3 \text{ kcal}\cdot\text{mol}^{-1}$), indicating a much more stronger adsorption strength between Th(IV) and

Py-TFIm-25 COF. Hence, when the Py-TFIm-25 COF is as the adsorbent, the Th(IV) ions are preferable to interact with the Py-TFIm-25 COF by occupying the adsorption sites, resulting in the significantly high adsorption selectivity towards Th(IV) for the adsorption experiments in this work. When multiple metal ions in solution compete for adsorption sites, the metal ions with a strong affinity for the adsorption site will preferentially occupy the adsorption site, preventing those with a relatively weak affinity from binding to the sites. This phenomenon is consistent with the previous report (*Chem. Eng. J.* **2018**, *344*, 594-603).

Comment 6: The source of the main materials should be marked.

Response: Thank you for your careful review and valuable advice. We have supplemented the sources of the main materials in the supplementary information of the revised manuscript.

Correction:

All starting reagents/reactants, except 1,3,6,8-tetra (4-aminophenyl) pyrene (PyTTA), 2',3',5',6'-tetrafluoro-[1,1':4',1''-terphenyl]-4,4''-dialdehyde (TFTDA), 5,6-bis(4-benzaldehyde)-1-methyl-benzimidazole (BFIm), and 5,6-bis(4-benzaldehyde)-1,3-dimethyl-benzimidazole-3-iodine salt (BFIm), were purchased commercially. 1,3,6,8-tetrabromopyrene was procured from Yanshen Technology Co., Ltd. 4-formylbenzeneboronic acid pinacol ester, camphorsulfonic, and 4,7-dibromo-2,1,3-benzothiadiazole were procured from Shanghai Adamas Reagent Co., Ltd. Tetrakis (triphenylphosphine) palladium (0), triethyl orthoformate, and 4-aminophenylboronic acid pinacol ester were procured from Meryer (Shanghai) Chemical Technology Co., Ltd. MeI was procured from Xiya Reagent. 1,4-dibromotetrafluorobenzene was procured from J&K Scientific. NaBH₄, anhydrous MgSO₄, anhydrous K₂CO₃, dioxane, ethyl acetate, tetrahydrofuran, ethanol, dichloromethane, and methanol were procured from Greagent of Shanghai Titan Scientific Co., Ltd. All reagents are of analytical grade and used directly as received without further purification.

(Page 3 in Revised supplementary information)

Special thanks for the constructive comments, which greatly helped us to improve the manuscript.

Reviewers' Comments:

Reviewer #1:

Remarks to the Author:

All my comments have been satisfactorily addressed. Publication is recommended.

Reviewer #2:

Remarks to the Author:

Liu et al. made commendable efforts to respond to our comments; however, we still find the manuscript insufficient for acceptance in Nature Communications. The main purpose of the paper is to map Thorium adsorption activity according to the electronic nature of different nitrogen functionalities. The authors' calculations show that the charge density of imidazole nitrogen is greater than that of imine and imidazolium, and they correlate this to the observed adsorption activity. Specifically, they attribute the greater capacity of Py-TFIm-25 to the presence of the neutral imidazole nitrogen. However, we believe that the experimental data do not support this claim. We think that the major adsorption sites are still the imine nitrogens, expressed by a significant capacity of the charged Py-TFIm-25 at high concentrations (>400 mg/g, Fig. S13). The reason why Py-TFIm-25 has a much greater maximum capacity (822 mg/g, Fig. S13) is because of its less severe positive charge – positive charge repulsion with thorium cations. Such repulsion becomes more significant when the pores are more packed with higher quantities of cations, provided by higher concentrations. This interaction also affects the observed adsorption kinetics. At low concentrations, the data show that the two materials perform quite similarly (Fig. 3a).

We have four main further comments on the manuscript:

Comment #1: The authors pointed out that at a similar concentration, the uptake capacity of Py-TFIm-25 COF is higher than that of COF-DL229. Although this is the observed data, it does not necessarily correlate to the higher affinity of imidazole nitrogen since it can still be due to the higher porosity of TFIm-25 COF. Furthermore, with the particular structures, the imine's electron density is lower than that of imidazole since it is highly deactivated by the electron-withdrawing effect of the fluorinated moiety. Therefore, it would be beneficial to have a non-fluorinated version of the series to have a fairer assessment of the relative affinity.

Comment #2: The authors explained their choice of the fluorinated TFTDA building block to enhance the interlayer interaction between the fluorinated moieties and non-fluorinated ones, which can lead to an improvement in crystallinity. However, the XRD spectra revealed an eclipsed AA stacking structure, which means the fluorinated benzene rings are right on top of each other. Does this contradict the stated electronic interaction? In our opinion, it would be more informative if the authors obtained non-fluorinated structures so that none of the nitrogen functionalities' electronics is compromised to have a more conclusive judgement.

Comment #3: The authors defended their claim of the role of imidazole nitrogen as the major binding force by presenting the increasing capacity gap between Py-TFImI-25 and Py-TFIm-25 when the initial thorium concentration is increased. We do not find this a convincing proof since when binding strength is under focus, sorbate concentration should be examined at low levels at which theoretical capacity (capacity at which all binding sites in a given structure reach their stoichiometric capture of the target) has not been reached to obtain an accurate insight into chemical interaction. Any excess capture beyond theoretical capacity must be the result of other secondary effects such as the reduction of the cations or interaction between adsorbed cations. For that, we would like to know the theoretical capacity of this series. Furthermore, the authors say that the data at pH 4 (bigger capacity gap) should be based on rather than those at pH 5 (little gap) since 1) at pH 5 and 25 ppm, the uptake in both materials are not at saturation points, and 2) at pH 4, more protonation would reduce the uptake

capability, leading to capacities obtained from the available nitrogen sites of the two materials (N-C=N- of Py-TFImI-25 COF vs NIm of Py-TFIm-25 COF). While the former statement may be true, the latter is not quite accurate since, given the higher charge density calculated for the Py-TFIm-25 imidazole nitrogen, this nitrogen site would be protonated before the imine nitrogens. Moreover, the majority functional group in both materials is still the imine. As a result, based on the collected data, we believe that the imine nitrogen is the main binding site.

Comment #4: The authors demonstrate the water stability of the imine COFs by showing the XRD patterns before and after treatment with water. However, we would like to know the mass recovery of the materials after the treatment. Did the authors recover the same quantity of materials after treatment as they started with? Also, the stability should be tested by dispersing the COF powder in aqueous solutions for prolonged time. Not one hour.

Reviewer #3:

Remarks to the Author:

The authors have noticed the inadequacies in the manuscript and have revised them carefully according to the questions and suggestions raised by the reviewers. I think the revised manuscript can now be accepted for publication in Nature Communications.

Response to reviewer #2:

Thank you for your professional comments on improving the quality of this manuscript, which also help us to understand more deeply. We have interpreted and revised the manuscript according to your comments. The revised parts are highlighted in red in the revised manuscript.

The point-to-point responses are summarized below:

Liu et al. made commendable efforts to respond to our comments; however, we still find the manuscript insufficient for acceptance in Nature Communications. The main purpose of the paper is to map Thorium adsorption activity according to the electronic nature of different nitrogen functionalities. The authors' calculations show that the charge density of imidazole nitrogen is greater than that of imine and imidazolium, and they correlate this to the observed adsorption activity. Specifically, they attribute the greater capacity of Py-TFIm-25 to the presence of the neutral imidazole nitrogen. However, we believe that the experimental data do not support this claim. We think that the major adsorption sites are still the imine nitrogens, expressed by a significant capacity of the charged Py-TFImI-25 at high concentrations (>400 mg/g, Fig. S13). The reason why Py-TFIm-25 has a much greater maximum capacity (822 mg/g, Fig. S13) is because of its less severe positive charge – positive charge repulsion with thorium cations. Such repulsion becomes more significant when the pores are more packed with higher quantities of cations, provided by higher concentrations. This interaction also affects the observed adsorption kinetics. At low concentrations, the data show that the two materials perform quite similarly (Fig. 3a).

Response: Thank you for your recognition of our work. We suppose that at pH 4, the imidazole N_{Im} sites of Py-TFIm-25 COF were not completely occupied by hydrogen ions, and the N_{Im} sites play an important role in the capture of thorium ions, as interpreted in Comment 3. For Py-TFImI-25 COF, the adsorption sites only the imine nitrogen sites. The similar Bader charge number of imine nitrogen sites for Py-TFImI-25 COF (-1.979) and Py-TFIm-25 COF (-1.983) indicated that the positive charge of the skeleton has little effect on the charge density of the imine nitrogen. Besides, the positively charged linkers only account for 1/4 of all the linkers. Therefore, we believe

that the positive charge – positive charge repulsion would not affect the capture capacity much. A more detailed explanation is provided in Comment 3.

Comment 1: The authors pointed out that at a similar concentration, the uptake capacity of Py-TFIm-25 COF is higher than that of COF-DL229. Although this is the observed data, it does not necessarily correlate to the higher affinity of imidazole nitrogen since it can still be due to the higher porosity of TFIm-25 COF. Furthermore, with the particular structures, the imine's electron density is lower than that of imidazole since it is highly deactivated by the electron-withdrawing effect of the fluorinated moiety. Therefore, it would be beneficial to have a non-fluorinated version of the series to have a fairer assessment of the relative affinity.

Response: Thank you for your careful review and for pointing this out. We agree that it is not available to conclude that imidazole nitrogen has higher affinity by comparing Py-TFIm-25 COF and COF-DL229, because the surface areas of Py-TFIm-25 COF and COF-DL229 differ significantly. But we can get such a conclusion by comparing Py-TFImI-25 COF and Py-TFIm-25 COF, because these two COFs were synthesized with only the linking units BFIm and BFImI different (Fig. R1), and have similar surface areas. In the linking unit BFIm, there are N_{Im} atoms that are not saturated with the -CH₃, which are new adsorption sites and enhanced the adsorption capacity.

Comparing the two structures of Py-TFImI-25 COF and COF-DL229 both with only imine sites, although the surface area of Py-TFImI-25 COF (1324.05 m² g⁻¹) is about twice that of COF-DL229 (569.30 m² g⁻¹), the nitrogen content of Py-TFImI-25 COF (3.84 mmol g⁻¹) is only about half of COF-DL229 (7.40 mmol g⁻¹). As such, Py-TFImI-25 COF and COF-DL229 exhibit comparable adsorption capacities.

We evaluated the non-fluorinated version of the series through DFT calculation. As shown in Table R1, the absolute values of Bader charge numbers of imidazole N_{Im} and imine N both increased slightly when the F atoms of the ligands were replaced by H atoms in both COFs. However, Py-TFImI-25 COF and Py-TFIm-25 COF were constructed using the same fluorine-containing ligand, and the relative positions of the fluorine-containing ligands in the two COFs were the same, so the electron-withdrawing effects of F atoms on N atoms of the two COFs were the same and would

not change the binding affinity order.

Fig. R1 The linking units BFIm of Py-TFImI-25 COF and BFIm of Py-TFIm-25 COF.

Table R1 Bader charge number (unit: $|e|$) of N sites in a series of configurations.

Configuration	N_{Im}	$N_{C=N}$
Py-TFImI-25-F	-	-1.979
Py-TFImI-25-H	-	-1.990
Py-TFIm-25-F	-2.165	-1.983
Py-TFIm-25-H	-2.191	-2.045

Comment 2: The authors explained their choice of the fluorinated TFTDA building block to enhance the interlayer interaction between the fluorinated moieties and non-fluorinated ones, which can lead to an improvement in crystallinity. However, the XRD spectra revealed an eclipsed AA stacking structure, which means the fluorinated benzene rings are right on top of each other. Does this contradict the stated electronic interaction? In our opinion, it would be more informative if the authors obtained non-fluorinated structures so that none of the nitrogen functionalities' electronics is compromised to have a more conclusive judgement.

Response: Thank you for your professional comment, which helped us to reconsider the structures of the two COFs. The two COFs are eclipsed AA stacking modes, but with the anti-isomer stacking mode, as shown in Fig. R2. The anti-isomer mode is more stable than the current syn-isomer mode due to the interlayer interaction (*J. Am. Chem. Soc.* **2013**, 135, 546-549). Since the adsorption sites are mainly located in the N atoms and not in the interlayer, the stacking of COFs in the z direction has little effect on the adsorption property. However, to ensure the rigorousness of the data, we

redid relevant quantitative calculations based on the anti-isomer mode, and the results showed that the calculation results based on the anti-isomer mode did not change much compared with the previous ones (Table R2). Corresponding amendments have been highlighted in red in the revised manuscript.

As interpreted in Comment 1, since the effect of F atoms on N atoms in both COFs is the parallel, we suppose that it would not affect the convincingness of the conclusion. As shown in Table R1, the affinity order of the N sites in the non-fluorinated structure is consistent with that of the fluorinated structures.

Fig. R2 The eclipsed AA stacking modes of the COFs

Table R2 The adsorption potential energies (ΔE) (kcal mol^{-1}) between metal cations and Py-TFImI-25 COF/Py-TFIm-25 COF adsorbents with anti-isomer mode.

COFs	Ions					
	Th(IV)	La(III)	Nd(III)	U(VI)	Sr(II)	Cs(I)
Py-TFImI-25 (syn-isomer)	-102.5	-48.0	-46.4	-15.6	-20.2	-9.2
Py-TFImI-25 (anti-isomer)	-106.9	-51.3	-47.1	-16.2	-22.6	-9.5
Py-TFIm-25 (syn-isomer)	-235.5	-90.3	-86.4	-35.8	-40.1	-20.1
Py-TFIm-25 (anti-isomer)	-248.6	-93.3	-87.2	-36.5	-40.9	-20.7

Comment 3: The authors defended their claim of the role of imidazole nitrogen as the major binding force by presenting the increasing capacity gap between Py-TFImI-25 and Py-TFIm-25 when the initial thorium concentration is increased. We do not find this a convincing proof since when binding strength is under focus, sorbate concentration should be examined at low levels at which theoretical capacity (capacity at which all binding sites in a given structure reach their stoichiometric capture of the target) has not been reached to obtain an accurate insight into chemical interaction. Any excess capture beyond theoretical capacity must be the result of other secondary effects such as the reduction of the cations or interaction between adsorbed cations. For that, we would like to know the theoretical capacity of this series. Furthermore, the authors say that the data at pH 4 (bigger capacity gap) should be based on rather than those at pH 5 (little gap) since 1) at pH 5 and 25 ppm, the uptake in both materials are not at saturation points, and 2) at pH 4, more protonation would reduce the uptake capability, leading to capacities obtained from the available nitrogen sites of the two materials ($N_{C=N}$ of Py-TFImI-25 COF vs N_{Im} of Py-TFIm-25 COF). While the former statement may be true, the latter is not quite accurate since, given the higher charge density calculated for the Py-TFIm-25 imidazole nitrogen, this nitrogen site would be protonated before the imine nitrogens. Moreover, the majority functional group in both materials is still the imine. As a result, based on the collected data, we believe that the imine nitrogen is the main binding site.

Response: Thank you for the thoughtful comment. As shown in Table R3, the theoretical capacities of Py-TFImI-25 COF and Py-TFIm-25 COF are 890.9 mg g⁻¹ and 1206.4 mg g⁻¹, respectively, based on the Th and N coordination ratio of 1:1 (Fig. 6a). The equilibrium adsorption capacities obtained so far are lower than the theoretical capacities.

Since the imidazole nitrogen site would be protonated before the imine nitrogen, we analyzed the occupation of the nitrogen sites at different thorium concentrations. When the initial concentration of Th(IV) is about 22 mg L⁻¹ and the pH is 4, the initial concentration is 0.1 mmol L⁻¹ for Th(IV) and 0.1 mmol L⁻¹ for H⁺, while the content of imidazole nitrogen (N_{Im}) is 0.35 mmol L⁻¹ (Table R4). Moreover, the hydrogen ions will

not be completely bound to imidazole nitrogen sites due to the presence of protonation constant. Therefore, at this concentration, the main adsorption site of Py-TFIm-25 COF is the imidazole nitrogen, while the adsorption site of Py-TFImI-25 COF is the imine nitrogen.

When the initial Th(IV) concentration increased 12 times, the uptake capacity of Py-TFImI-25 COF for Th(IV) increased by 10.9 times and the uptake capacity of Py-TFIm-25 COF for Th(IV) increased by 13.4 times (Table R4 and Table R5). In this adsorption system, there are hydrolysis equilibriums of thorium ions, proton equilibriums, and reaction equilibriums. When the pH is fixed, the difference between Py-TFImI-25 COF and Py-TFIm-25 COF adsorption systems should focus on the difference of the reaction equilibrium constants, k_1 for the imine site and k_2 for the imidazole site. The 10.9-fold increase in the uptake capacity of Py-TFImI-25 COF is due to the contribution of k_1 , while the 13.4-fold increase in the uptake capacity of Py-TFIm-25 COF is due to the combined contribution of k_1 and k_2 . Therefore, at higher concentrations, the contribution of imidazole nitrogen cannot be neglected.

Table R3 The theoretical capacity of Py-TFImI-25 COF and Py-TFIm-25 COF.

COFs	N _{C=N} content (mmol g ⁻¹)	N _{Im} content (mmol g ⁻¹)	N _{Im-CH₃} content (mmol g ⁻¹)	Total N content (mmol g ⁻¹)	Theoretical capacity (mg g ⁻¹)
Py-TFImI-25	3.84	0	1.92	5.76	890.9
Py-TFIm-25	4.16	1.04	1.04	6.24	1206.4

Table R4 Analysis of nitrogen sites (Solid-liquid ratio = 1:3000 g mL⁻¹, Initial concentration of Th(IV) \approx 22 mg L⁻¹, pH = 4, T= 25°C).

COFs	N _{C=N} content (mmol L ⁻¹)	N _{Im} content (mmol L ⁻¹)	Th(IV) concentration (mmol L ⁻¹)	H ⁺ concentration (mmol L ⁻¹)	Th(IV) uptake capacity (mg g ⁻¹)
Py-TFImI-25	1.28	0	0.1	0.1	43.1
Py-TFIm-25	1.39	0.35	0.1	0.1	61.2

Table R5 Analysis of nitrogen sites (Solid-liquid ratio = 1:3000 g mL⁻¹, Initial concentration of Th(IV) \approx 273 mg L⁻¹, pH = 4, T= 25°C).

COFs	N _{C=N} content (mmol L ⁻¹)	N _{Im} content (mmol L ⁻¹)	Th(IV) concentration (mmol L ⁻¹)	H ⁺ concentration (mmol L ⁻¹)	Th(IV) uptake capacity (mg g ⁻¹)
Py-TFImI-25	1.28	0	1.2	0.1	468.3
Py-TFIm-25	1.39	0.35	1.2	0.1	822.6

Comment 4: The authors demonstrate the water stability of the imine COFs by showing the XRD patterns before and after treatment with water. However, we would like to know the mass recovery of the materials after the treatment. Did the authors recover the same quantity of materials after treatment as they started with? Also, the stability should be tested by dispersing the COF powder in aqueous solutions for prolonged time. Not one hour.

Response: Thank you for your professional comments. Unfortunately, we were unable to obtain the mass recovery due to the unavoidable loss of samples during the transfer process. However, we hope that the N content in water before and after water treatment could indicate the stability of the two COFs. The N content before and after water treatment was carried out on HQ40D-Multi (Hach). As shown in Table R6, the loss of both COFs is less than 1.5%, indicating good water stability. Since all adsorption experiments were completed within 1 h, we choose 1 h to test the water stability. Meanwhile, 6 h was set to test the water stability. As shown in Fig. R3, both materials exhibited good stability after soaking in water and maintained the integrity of the skeleton and the crystal structure.

Table R6 The loss of Py-TFImI-25 COF and Py-TFIm-25 COF after treatment with water (Solid-liquid ratio = 1:3000 g mL⁻¹, Shaking time = 6 h, T= 25°C).

COFs	Initial Concentration of COFs suspension (mg L ⁻¹)	N content in COFs (mmol g ⁻¹)	N content in water before treatment (mg L ⁻¹)	N content after 6 h treatment (mg L ⁻¹)	The loss of COFs in water after 6 h treatment (mg L ⁻¹)	The Loss of COFs
Py-TFImI-25	333.33	5.76	0	0.1	1.24	0.37%
Py-TFIm-25	333.33	6.24	0	0.4	4.58	1.37%

Fig. R3 PXRD patterns of Py-TFImI-25 COF and Py-TFIm-25 COF after soaking in water. (Solid-liquid ratio = 1:3000 g mL⁻¹, Shaking time = 1 h and 6 h, T= 25°C).

We appreciate your professional work earnestly. Thank you very much for your comments and suggestions.

Reviewers' Comments:

Reviewer #2:

Remarks to the Author:

The authors have done further work on the matter of imine-imidazole binding modes for the metal capture. Although it is still not completely clear to us, we'd like to take a leap of faith and let the community decide. There is ample data now and should make a story. The new data should also be presented in the SI since it will be published along with reviews anyway.

Response to reviewer #2:

The authors have done further work on the matter of imine-imidazole binding modes for the metal capture. Although it is still not completely clear to us, we'd like to take a leap of faith and let the community decide. There is ample data now and should make a story. The new data should also be presented in the SI since it will be published along with reviews anyway.

Response: Thank you for your recognition of our work. We have supplemented the new data in the Supplementary Information.

Thanks again for your professional comments and suggestions.